# Hybrid Search for Efficient Planning with Completeness Guarantees

**Kalle Kujanpää**[1,3],[*] **Joni Pajarinen**[2,3], **Alexander Ilin**[1,3,4]

[1]Department of Computer Science, Aalto University
[2]Department of Electrical Engineering and Automation, Aalto University
[3]Finnish Center for Artificial Intelligence FCAI
[4]System 2 AI
`{kalle.kujanpaa,joni.pajarinen,alexander.ilin}@aalto.fi`

## Abstract

Solving complex planning problems has been a long-standing challenge in computer science. Learning-based subgoal search methods have shown promise in tackling these problems, but they often suffer from a lack of completeness guarantees, meaning that they may fail to find a solution even if one exists. In this paper, we propose an efficient approach to augment a subgoal search method to achieve completeness in discrete action spaces. Specifically, we augment the high-level search with low-level actions to execute a multi-level (hybrid) search, which we call *complete subgoal search*. This solution achieves the best of both worlds: the practical efficiency of high-level search and the completeness of low-level search. We apply the proposed search method to a recently proposed subgoal search algorithm and evaluate the algorithm trained on offline data on complex planning problems. We demonstrate that our complete subgoal search not only guarantees completeness but can even improve performance in terms of search expansions for instances that the high-level could solve without low-level augmentations. Our approach makes it possible to apply subgoal-level planning for systems where completeness is a critical requirement.

## 1 Introduction

Combining planning with deep learning has led to significant advances in many fields, such as automated theorem proving [33], classical board games [39], puzzles [1], Atari games [38], video compression [24], robotics [5], and autonomous driving [43]. However, deep learning-based methods often plan in terms of low-level actions, such as individual moves in chess or commands in Atari games, whereas humans tend to plan by decomposing problems into smaller subproblems [11]. This observation has inspired a lot of recent research into methods that solve complex tasks by performing hierarchical planning and search. In the continuous setting, hierarchical planning has been applied to, for instance, robotic manipulation and navigation [31, 25, 12, 6, 15]. Discrete hierarchical planning methods, when trained entirely on offline data, can solve complex problems with high combinatorial complexity. These methods are efficient at long-term planning thanks to the hierarchy reducing the effective planning horizon [4, 18, 45]. Furthermore, they reduce the impact of noise on planning [4] and show promising generalization ability to out-of-distribution tasks [45]. Training RL agents on diverse multi-task offline data has been shown to scale and generalize broadly to new tasks, and being compatible with training without any environment interaction can be seen as an additional strength of these methods [20, 5, 42].

---

[*]Corresponding author

37th Conference on Neural Information Processing Systems (NeurIPS 2023).

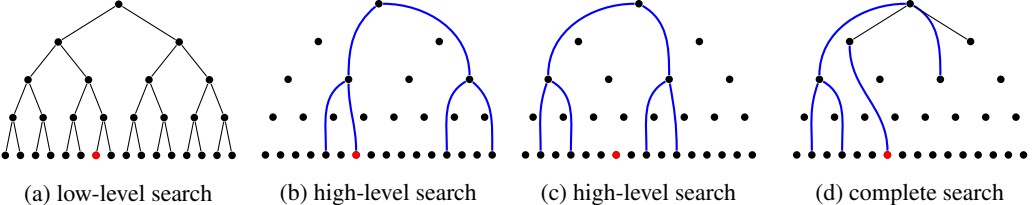

| (a) low-level search | (b) high-level search | (c) high-level search | (d) complete search |

Figure 1: (a): Low-level search systematically visits all states reachable from the root node and, therefore, is guaranteed to find a path to the terminal state (shown in red). (b): High-level search can find a solution with much fewer steps due to its ability to operate with high-level actions that span several time steps. (c): High-level search may fail to find a solution due to an imperfect subgoal generation model. (d): Complete subgoal search can find a solution with few steps thanks to high-level actions, and it has completeness guarantees due to also considering low-level actions.

Despite their excellent performance, the discrete subgoal search methods share a weakness: neither AdaSubS [45], kSubS [4], nor HIPS [18] are guaranteed to find a solution even if it exists. All these methods rely on a learned subgoal generator that proposes new subgoals to be used with some classical search algorithms for planning. If the generator fails to perform adequately, the result can be a failure to discover solutions to solvable problems. We call this property the lack of completeness. The guarantee of the algorithm finding a solution is critical for both theoretical science and practical algorithms. In the context of subgoal search, completeness guarantees discovering the solution, even if the generative model is imperfect. Completeness allows extensions and new incremental algorithms that require building on top of exact solutions. One example of that could be curriculum learning. Furthermore, we show that completeness significantly improves the out-of-distribution generalization of subgoal search. Achieving completeness also makes applying high-level search as an alternative to low-level search possible in safety-critical real-world systems. Zawalski et al. [45] mentioned that AdaSubS can be made complete by adding an exhaustive one-step subgoal generator that is only utilized when the search would otherwise fail, whereas Kujanpää et al. [18] proposed combining high- and low-level actions to attain completeness of HIPS. However, to the best of our knowledge, this idea has not been formalized, analyzed, or evaluated in prior work.

We present a multi-level (hybrid) search method, *complete subgoal search*, that combines hierarchical planning and classical exhaustive low-level search. The idea is illustrated in Figure 1. We apply our complete search approach to HIPS [18] which 1) has been shown to outperform several other subgoal search algorithms in challenging discrete problem domains, 2) does not require training multiple subgoal generators in parallel to perform adaptive-length planning, and 3) is particularly suited for being combined with low-level actions due to using Policy-guided Heuristic Search (PHS) as the main search algorithm [18, 27]. We use the name HIPS-$\varepsilon$ for the proposed enhancement of HIPS with complete subgoal search. We argue that the idea suggested in [45] can be seen as a specific case of our search method that works well in some environments but can be improved upon in others. We evaluate HIPS-$\varepsilon$ on challenging, long-term discrete reasoning problems and show that it outperforms HIPS, other subgoal search methods, and strong offline reinforcement learning (RL) baselines. Not only does HIPS-$\varepsilon$ attain completeness, but our complete subgoal search also demonstrates improved performance on problem instances that subgoal search alone was sufficient for solving.

## 2 Related Work

We propose to augment the high-level search used in learning-based hierarchical planning with low-level actions to achieve completeness while even improving the search performance in terms of node expansions. First, we discuss our work in the general context of hierarchical planning and then focus on the closest methods in discrete subgoal search, to which we compare the proposed approach.

**Continuous Hierarchical Planning** Hierarchical planning has been widely applied in the continuous setting. The Cross-Entropy Method (CEM) [34] is often used as the planning algorithm for different hierarchical planning methods [22, 25, 12, 32]. However, the cross-entropy method is a numerical optimization method, unsuitable for addressing discrete planning problems. There is also prior work on hierarchical planning for continuous and visual tasks, including control and navigation,

where CEM is not used as the optimization algorithm [21, 26, 6, 31, 36, 46]. Kim et al. [14] plan how to select landmarks and use them to train a high-level policy for generating subgoals. Planning with search is a common approach in classical AI research [35]. Combining search with hierarchical planning has been proposed [23, 8], but these approaches are generally inapplicable to complex, discrete reasoning domains, where precise subgoal generation and planning are necessary [45].

**Recursive Subgoal-Based Planning**   In this work, we focus on search algorithms where subgoals are generated sequentially from the previous subgoal. Recursive subgoal generation is an orthogonal alternative to our approach. In certain domains, given a start and a goal state, it is possible to hierarchically partition the tasks into simpler ones, for instance, by generating subgoals that are approximately halfway between the start and the end state. This idea can be applied repeatedly for an efficient planning algorithm [30, 31, 13]. However, the recursive approach can be difficult to combine with search and limits the class of problems that can be solved.

**Discrete Subgoal Search**   Discrete subgoal search methods can solve complex reasoning problems efficiently with limited search node expansions. S-MCTS [7] is a method for MCTS-based subgoal search with predefined subgoal generators and heuristics, which significantly limits its usability to novel problems. Allen et al. [2] improve the efficiency of search through temporal abstraction by learning macro-actions, which are sequences of low-level actions. Methods that learn subgoal generators, on the other hand, generally suffer from a lack of completeness guarantees. kSubS [4] learns a subgoal generator and performs planning in the subgoal space to solve demanding reasoning tasks. AdaSubS [45] builds on top of kSubS and proposes learning multiple subgoal generators for different distances and performing adaptive planning. HIPS [18] learns to segment trajectories using RL and trains one generator that can propose multiple-length subgoals for adaptive planning.

# 3   Method

We present HIPS-$\varepsilon$, an extension of *Hierarchical Imitation Planning with Search* (HIPS), a search-based hierarchical imitation learning algorithm for solving difficult goal-conditioned reasoning problems [18]. We use the Markov decision process (MDP) formalism, and the agent's objective is to enter a terminal state in the MDP. However, we deviate from the reward maximization objective. Instead, the objective is to minimize the total search loss, that is, the number of search nodes expanded before a solution is found. A similar objective has been adopted in prior work on subgoal search and, in practice, the solutions found are efficient [4, 45].

We assume the MDPs to be deterministic and fully observable with a discrete state space $\mathcal{S}$ and action space $\mathcal{A}$. We also assume the standard offline setting: the agent cannot interact with the environment during training, and it learns to solve tasks only from $\mathcal{D}$, an existing dataset of expert demonstrations. These trajectories may be heavily suboptimal, but all lead to a goal state. Therefore, the agent should be capable of performing stitching to discover the solutions efficiently [40].

## 3.1   Preliminaries

HIPS learns to segment trajectories into subgoals using RL and a subgoal-conditioned low-level policy $\pi(a|s, s_g)$. The segmented trajectories are used to train a generative model $p(s_g|s)$ over subgoals $s_g$ that is implemented as a VQVAE with discrete latent codes [41], a strategy inspired by Ozair et al. [29]. The VQVAE learns a state-conditioned prior distribution $p(e|s)$ over latent codes $e$ from a codebook $\mathbb{E}$ and a decoder $g(s_g|e, s)$ acting as a generator that outputs a subgoal $s_g$ given the latent code $e$ and state $s$ deterministically as $s_g = g(e, s)$. Each code $e_k \in \mathbb{E}$ can be considered a state-dependent high-level action that induces a sequence of low-level actions $A_k(s) = (a_1, \ldots, a_{n_k})$, $a_i \in \mathcal{A}$, that takes the environment from state $s$ to $s_g$. The low-level action sequence is generated deterministically by the subgoal-conditioned policy $\pi(a|s, s_g)$, when the most likely action is used at each step. The prior $p(e|s)$ can be interpreted as a high-level policy $\pi_{\text{SG}}(e|s)$ that assigns a probability to each latent code. HIPS also learns a heuristic that predicts the number of low-level actions $a \in \mathcal{A}$ needed to reach a terminal state from $s$ and a single-step dynamics model $f_{\text{dyn}}(s_{i+1}|a_i, s_i)$. The heuristic is denoted by $V(s)$ in [18], but we denote it by $h$ in this paper.

HIPS performs high-level planning in the subgoal space with PHS [27], although other search methods can be used as well. PHS is a variant of best-first search in which the algorithm maintains

a priority queue and always expands the node $n$ with the lowest evaluation function value. There is a non-negative loss function $L$, and when the search expands node $n$, it incurs a loss $L(n)$. The objective of PHS is to minimize the total *search* loss. The evaluation function of PHS is

$$\varphi(n) = \eta(n)\frac{g(n)}{\pi(n)}, \tag{1}$$

where $\eta(n) \geq 1$ is a heuristic factor that depends on the heuristic function $h(n)$. $g(n)$ is the *path loss*, the sum of the losses from the root $n_0$ to $n$. $\pi(n) \leq 1$ is the probability of node $n$ that is defined recursively: $\pi(n') = \pi(n'|n)\pi(n), \pi(n_0) = 1$, $n$ is the parent of $n'$ and $n \in \text{desc}_*(n_0)$, that is, $n$ is a descendant of the root node $n_0$. When HIPS plans, each search node $n$ corresponds to a state $s$, $g(n)$ to the number of low-level steps needed to reach $n$ from the root $n_0$, and $h(n)$ to the output of the value function $V(s)$. The children $\mathcal{C}(n)$ of node $n$ are the valid subgoals generated by the VQVAE decoder $g(e_k, s)$ for all $e_k \in \mathbb{E}$. The search policy $\pi(n)$ is induced by the conditional policy $\pi(n'|n)$, which is represented by the state-conditioned prior $p(e|s)$.

## 3.2 Complete Subgoal Search

HIPS performs search solely in the space of subgoals proposed by the trained generator network, which may cause failures to discover solutions to solvable problems due to possible imperfections of the subgoal generation model (see Fig. 1c). This problem can be tackled in discrete-action MDPs by augmenting the search with low-level actions (see Fig. 1d). In this work, we apply this idea to HIPS to guarantee solution discovery when the solution exists while still utilizing the subgoal generator for temporally abstracted efficient search.

Formally, we propose modifying the search procedure of HIPS such that in addition to the subgoals $\{s_H = g(e, s), \forall e \in \mathbb{E}\}$ produced by the HIPS generator, the search also considers states $\{s_L = T(s, a), \forall a \in \mathcal{A}\}$ that result from the agent taking every low-level action $a \in \mathcal{A}$ in state $s$, where $T(s, a)$ is the environment transition function. We assume that $T$ is known but we also evaluate the method with a learned $T$. The augmented action space is then $\mathcal{A}^+ = \mathcal{A} \cup \mathbb{E}$. We use PHS as the search algorithm and compute the node probabilities used for the evaluation function (1) as

$$\pi(n) = \prod_{j=1}^{d} \pi\left(s_j(n)|s_{j-1}(n)\right),$$

where $s_d(n)$ is the state that corresponds to node $n$ at depth $d$, $s_{d-1}(n), s_{d-2}(n), ..., s_0(n)$ are the states of the ancestor nodes of $n$ and $\pi(s_j(n)|s_{j-1}(n))$ is the search policy. We propose to compute the probabilities $\pi(s_j|s_{j-1})$ in the following way:

$$\pi(s_j|s_{j-1}) = \begin{cases} (1-\varepsilon)\pi_{\text{SG}}(e_k|s_{j-1}) & \text{if } s_j = g(e_k, s_{j-1}) \text{ is proposed by HIPS} \\ \varepsilon\,\pi_{\text{BC}}(a|s_{j-1}) & \text{if } s_j = T(s_{j-1}, a) \text{ is proposed by low-level search} \end{cases} \tag{2}$$

where $\pi_{\text{SG}}(e_k|s_{j-1}) = p(e_k|s_{j-1})$ is the prior over the high-level actions learned by HIPS and $\pi_{\text{BC}}(a|s_{j-1})$ is a low-level policy that we train from available expert demonstrations with BC. We use hyperparameter $\varepsilon$ to balance the probabilities computed with the high and low-level policies: higher values of $\varepsilon$ prioritize more low-level exploration. We call this complete search HIPS-$\varepsilon$.

The proposed complete search approach can be combined with any policy-guided search algorithm, but we focus on PHS due to its state-of-the-art performance. Complete search can also be used with policyless search algorithms, but in that case, $\varepsilon$ cannot directly control the frequency of low and high-level actions.

Zawalski et al. [45] suggested using low-level actions only when the high-level search would otherwise fail. In the context of PHS, this can be interpreted as having an infinitesimal $\varepsilon$ such that every high-level action has a lower evaluation function value (1) and, therefore, a higher priority than every low-level action. With some abuse of notation, we refer to this approach as $\varepsilon \to 0$.

## 3.3 Complete Subgoal Search Heuristic

The objective of PHS is to minimize the search loss, and Orseau and Lelis [27] propose using a heuristic to estimate the $g$-cost of the least costly solution node $n^*$ that is a descendant of the current

search node $n$. They propose to approximate the probability $\pi(n^*)$ for node $n^*$ that contains a found solution as

$$\pi(n^*) = [\pi(n)^{1/g(n)}]^{g(n)+h(n)} \tag{3}$$

and use this approximation to arrive at a heuristic factor $\hat{\eta}_h(n) = \frac{1+h(n)/g(n)}{\pi(n)^{h(n)/g(n)}}$ to be used in (1). PHS* is the variant of PHS that uses this heuristic factor. The derivation of the heuristic factor $\hat{\eta}_h(n)$ assumes that $g(n)$, the path loss of node $n$, and $h(n)$, the heuristic function approximating the distance to the least-cost descendant goal node $n^*$ are expressed in the same scale. This does not hold for the proposed search algorithm. The search objective of HIPS-$\varepsilon$ is to minimize the number of node expansions, which implies that $g(n)$ is equal to the depth of node $n$, whereas the heuristic $h(n)$ is trained to predict the number of low-level actions to the goal. One solution is to let $g(n)$ be equal to the number of low-level steps to node $n$. However, this would distort the search objective. The cost of expanding a node corresponding to a subgoal would equal the length of the corresponding low-level action sequence, whereas the loss of low-level actions would be one. However, the cost should be the same for every node, as the objective is to minimize the number of node expansions.

Instead, we propose re-scaling the heuristic factor $h$ to be equal to the estimated number of search nodes on the path from $n$ to the goal node $n^*$ by dividing the expected number of low-level actions from node $n$ to the terminal node by the average low-level distance between the nodes on the path from $n_0$ to node $n$. Let $g(n)$ be equal to $d(n)$, the depth of the node, and let $l(n)$ be the number of low-level actions from the root node. Then, we define a scaled heuristic $\hat{h}(n) = \frac{h(n)g(n)}{l(n)}$ and by using it instead of $h(n)$ in (3), we get a new approximation $\pi(n^*) = \pi^{(1+h(n)/l(n))}$, which yields the following heuristic factor and evaluation function:

$$\hat{\eta}_{\hat{h}} = \frac{1 + h(n)/l(n)}{\pi(n)^{h(n)/l(n)}}, \qquad \hat{\varphi}_{\hat{h}}(n) = \frac{g(n) \cdot (1 + h(n)/l(n))}{\pi(n)^{1+h(n)/l(n)}}. \tag{4}$$

For the full derivation, please see Appendix C. Note that $\hat{\eta}_{\hat{h}}$ may not be PHS-admissible, even if $h(n)$ were admissible. In HIPS, there are no guarantees of the learned heuristic $h(n)$ being admissible.

### 3.4 Analysis of HIPS-$\varepsilon$

Orseau and Lelis [27] showed that the search loss of PHS has an upper bound:

$$L(\text{PHS}, n^*) \leq L_{\text{U}}(\text{PHS}, n^*) = \frac{g(n^*)}{\pi(n^*)} \eta^+(n^*) \sum_{n \in \mathcal{L}_\varphi(n^*)} \frac{\pi(n)}{\eta^+(n)}, \tag{5}$$

where $n^*$ is the solution node returned by PHS, $g(n)$ is the path loss from the root to $n$, $\pi$ the search policy, and $\eta^+$ the modified heuristic factor corresponding to the monotone non-decreasing evaluation function $\varphi^+(n)$, and $\mathcal{L}_\varphi(n^*)$ the set of nodes that have been expanded before $n^*$, but the children of which have not been expanded. Now, given the recursively defined policy $\pi(n)$ that depends on $\pi_{\text{BC}}$ and $\pi_{\text{SG}}$ as outlined in Subsection 3.2, the upper bound derived in (5) is unchanged.

However, the upper bound of (5) can be seen as uninformative in the context of complete search, as it depends on the high-level policy $\pi_{\text{SG}}$. In general, we cannot make any assumptions about the behavior of the high-level policy as it depends on the subgoals proposed by the generator network. However, we can derive the following upper bound, where the probability of the terminal node $n^*$ is independent of the behavior of the generator, and consequently, the high-level policy $\pi_{\text{SG}}$:

**Corollary 1.** *(Subgoal-PHS upper bound without $\pi_{SG}$ in the denominator). For any non-negative loss function $L$, non-empty set of solution nodes $\mathcal{N}_\mathcal{G} \subseteq desc_*(n_0)$, low-level policy $\pi_{BC}$, high-level policy $\pi_{SG}$, $\varepsilon \in (0, 1]$, policy $\pi(n'|n)$ as defined in (2), and heuristic factor $\eta(\cdot) \geq 1$, Subgoal-PHS returns a solution node $n^* \in \arg\min_{n^* \in \mathcal{N}_\mathcal{G}} \varphi^+(n^*)$. Then, there exists a terminal node $\hat{n} \in \mathcal{N}_\mathcal{G}$ such that $state(\hat{n}) = state(n^*)$, where $\hat{n}$ corresponds to low-level trajectory $(s_0, a_0, \ldots, a_{N-1}, state(n^*))$ and the search loss is bounded by*

$$L(\text{PHS-h}, n^*) \leq L(\text{PHS-h}, \hat{n}) \leq \frac{g(\hat{n})}{\varepsilon^N \prod_{i=0}^{N-1} \pi_{BC}(a_i|s_i)} \eta^+(\hat{n}) \sum_{n \in \mathcal{L}_\varphi(\hat{n})} \frac{\pi(n)}{\eta^+(n)} \tag{6}$$

*Proof.* Subgoal-PHS does not change the underlying search algorithm, so the assumption of Subgoal-PHS returning the minimum score solution node follows from Theorem 1 in [27]. The first inequality

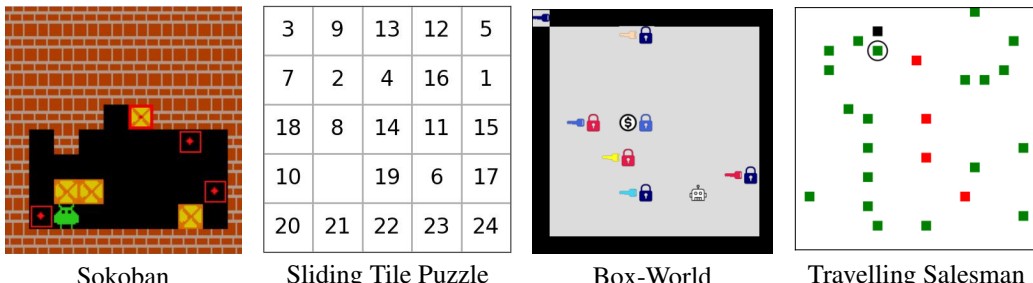

Figure 2: We evaluate our complete search approach, HIPS-$\varepsilon$, in Sokoban, Sliding Tile Puzzle (STP), Box-World (BW), and Travelling Salesman Problem (TSP). In Sokoban, the agent must push the yellow boxes onto the red target locations. In the Sliding Tile Puzzle, the agent must slide the tiles to sort them from 1 to 24. In Box-World, the task is to collect the gem (marked with $) by opening locks with keys of the corresponding color. In Travelling Salesman, the agent (marked with a circle) must visit all unvisited cities (red squares) before returning to the start (black square).

follows from the fact that the loss function $L$ is always non-negative and the node $n^*$ will be expanded before any other node $n$ with $\varphi^+(n) > \varphi^+(n^*)$. In particular, note that it is possible that $n^* = \hat{n}$. The second inequality follows from the Theorem 1 in [27] and $\pi(\hat{n}) = \varepsilon^N \prod_{i=0}^{N-1} \pi_{\text{BC}}(a_i|s_i)$. $\quad\square$

Now, suppose that we have a problem where there is only one terminal state and sequence of low-level actions that reaches this terminal state, and the search space $\mathcal{S}$ is infinite. Let us denote this trajectory as $\mathcal{T} = (s_0, a_0, s_1, a_1, \ldots, s_{N-1}, a_{N-1}, s_N)$. Additionally, assume that the high-level policy $\pi_{\text{SG}}$ never guides the search towards a solution, that is, for each state $s \in \mathcal{T}$, the low-level action sequence for every subgoal $s_g$ proposed by the generator network contains an incorrect action. Furthermore, assume that the generator network can always generate new, reachable subgoals for all states $s \in \mathcal{S}$. Hence, only actions selected by the low-level policy lead to the solution, and thus $\pi(n^*) = \varepsilon^N \prod_{i=0}^{N-1} \pi_{\text{BC}}(a_i|s_i)$. If $\varepsilon = 0$ or $\varepsilon \to 0$, the search does not converge, and the upper bound in (5) approaches infinity.

Czechowski et al. [4] argue that low-level search suffers more from local noise than subgoal search and using subgoals improves the signal-to-noise ratio of the search, and the results of Kujanpää et al. [18] support this. For complete search, a lower $\varepsilon$ can increase the use of high-level policy in relation to the low-level actions, thereby shortening the effective planning horizon and making the search more effective. Hence, we hypothesize that a low value of $\varepsilon$ is generally preferred. However, (6) shows that the worst-case performance of the search deteriorates when the value of $\varepsilon$ decreases.

If there is no heuristic, the bound can be simplified similarly as in Orseau and Lelis [27]. This following corollary shows that removing the heuristic factor does not affect the inverse dependence of the worst-case search loss on $\varepsilon$, and in fact, the worst case of (6), that is, the search failing for $\varepsilon \to 0$, is possible even given a perfect heuristic function.

**Corollary 2.** *(Subgoal-PHS upper bound without heuristic). If there is no heuristic, that is, $\forall n, \eta(n) = 1$, the upper bound simplifies to*

$$L(\text{PHS-h}, n^*) \le L(\text{PHS-h}, \hat{n}) \le \frac{g(\hat{n})}{\varepsilon^N \prod_{i=0}^{N-1} \pi_{BC}(a_i|s_i)} \tag{7}$$

*Proof.* The first inequality follows as in Corollary 1, and the second inequality follows from the assumptions and $\sum_{n' \in \mathcal{L}_\varphi(n)} \pi(n') \le 1$ for all $n$, see Orseau et al. [28]. $\quad\square$

## 4 Experiments

The main objective of our experiments is to evaluate whether the proposed complete search algorithm HIPS-$\varepsilon$ can be combined with an existing subgoal search algorithm, HIPS [18], to improve its performance in environments that require long-term planning and complex, object-based relational reasoning. We also analyze the sensitivity of HIPS-$\varepsilon$ to the hyperparameter $\varepsilon$ and the impact of the

Table 1: The success rates (%) after performing $N$ node expansions for different subgoal search algorithms with access to environment dynamics. For HIPS-$\varepsilon$, we use the value of $\varepsilon$ that yields in the best performance: $\varepsilon \to 0$ for Sokoban, $\varepsilon = 10^{-5}$ for Sliding Tile Puzzle, $\varepsilon = 10^{-3}$ for Box-World and $\varepsilon \to 0$ for Travelling Salesman Problem. HIPS corresponds to HIPS-env in [18] and uses PHS* as the search algorithm in all environments.

|  | Sokoban | | | | Sliding Tile Puzzle | | | |
|---|---|---|---|---|---|---|---|---|
| $N$ | 50 | 100 | 200 | $\infty$ | 50 | 100 | 200 | $\infty$ |
| PHS* (low-level search) | 0.2 | 2.4 | 16.2 | **100** | 0.0 | 0.0 | 0.0 | **100** |
| HIPS (high-level search) | 82.0 | 87.8 | 91.6 | 97.9 | 8.7 | 56.8 | 86.3 | 95.0 |
| AdaSubS (high-level search) | 76.4 | 82.2 | 85.7 | 91.3 | 0.0 | 0.0 | 0.0 | 0.0 |
| kSubS (high-level search) | 69.1 | 73.1 | 76.3 | 90.5 | 0.7 | **79.9** | 89.8 | 93.3 |
| HIPS-$\varepsilon$ (complete search) | **84.3** | **89.5** | **93.1** | **100** | **18.5** | 69.5 | **93.8** | **100** |
|  | Box-World | | | | Travelling Salesman Problem | | | |
| $N$ | 5 | 10 | 30 | $\infty$ | 20 | 50 | 100 | $\infty$ |
| PHS* (low-level search) | 0.0 | 0.1 | 2.2 | **100** | 0.0 | 0.0 | 0.0 | **100** |
| HIPS (high-level search) | 86.3 | 97.9 | 99.9 | 99.9 | **19.6** | **88.1** | 97.7 | **100** |
| AdaSubS (high-level search) |  |  |  |  | 0.0 | 0.0 | 0.6 | 21.2 |
| kSubS (high-level search) |  |  |  |  | 0.0 | 1.5 | 40.4 | 87.9 |
| HIPS-$\varepsilon$ (complete search) | **89.7** | **98.9** | **100** | **100** | 17.9 | 87.4 | **97.9** | **100** |

novel heuristic factor and the corresponding PHS* evaluation function (4). The OOD-generalization abilities of HIPS-$\varepsilon$ are also discussed. We use the four environments considered in [18]: Box-World [44], Sliding Tile Puzzle [17], Gym-Sokoban [37] and Travelling Salesman (see Fig. 2).

We implement HIPS-$\varepsilon$ as proposed in (2) and the policy $\pi_{\text{BC}}(a|s)$ as a ResNet-based network [10] and train it using Adam [16]. We use the implementation of HIPS from the original paper and use PHS* (or our complete variant thereof) as the search algorithm for HIPS and HIPS-$\varepsilon$. In [18], HIPS was evaluated with GBFS and A* on some environments. HIPS-$\varepsilon$ experiments with these search algorithms can be found in Appendix F. We use the versions of HIPS-$\varepsilon$ and HIPS that have access to the environment dynamics (named HIPS-env in [18]) unless stated otherwise.

In Table 1, we report the proportion of solved puzzles as a function of the number of expanded nodes for the low-level search PHS*, high-level search algorithms HIPS [18], kSubS [4], and AdaSubS [45], and our HIPS-$\varepsilon$, which is HIPS enhanced with the proposed complete subgoal search. We used the kSubS and AdaSubS results from [18] where they were not evaluated on Box-World. The results show that high-level search alone does not guarantee completeness, which means that for some puzzles the solution is not found even without a limit on the number of expanded nodes ($N = \infty$) due to the generator failing to propose subgoals that lead to a valid terminal state, a phenomenon observed in prior work [4, 18, 45]. As expected, HIPS-$\varepsilon$ achieves a 100% success rate in all environments thanks to augmenting the subgoals with low-level actions. In addition to the number of search node expansions, we analyzed the search cost in terms of wall-clock time and environment steps. HIPS-$\varepsilon$ generally outperforms the baselines also using these metrics. The results of these experiments can be found in Appendices J and K.

We also evaluated HIPS-$\varepsilon$ on Sokoban, STP, and TSP with learned transition models and compared it to HIPS with learned models, behavioral cloning (BC), Conservative Q-learning [CQL, 19], and Decision Transformer [DT, 3]. We omitted Box-World due to HIPS struggling with the learned dynamics [18]. The results are shown in Table 2. HIPS-$\varepsilon$ outperforms the baselines but can fail to find a solution to a solvable problem instance due to the learned dynamics model sometimes outputting incorrect transitions. Completeness can be attained by using the true dynamics for validating solutions and simulating low-level actions, while still minimizing the number of environment steps (see Appendix E). For complete results with confidence intervals, please see Appendix G.

Interestingly, the results in Table 1 show that augmenting the high-level search with low-level actions may have a positive impact on the performance for the same budget of node expansions $N$: we observe statistically significantly increased success rates for the same $N$ in all environments except

Table 2: The success rates (%) of different algorithms without access to environment dynamics in Sokoban, Sliding Tile Puzzle, and TSP. HIPS-$\varepsilon$ outperforms the baselines and can solve 100% of the puzzles when the environment dynamics are easy to learn.

|  | HIPS | HIPS-$\varepsilon$ | BC | CQL | DT |
|---|---|---|---|---|---|
| Sokoban | 97.5 | **100.0** | 18.7 | 3.3 | 36.7 |
| Sliding Tile Puzzle | 94.7 | **100.0** | 82.5 | 11.7 | 0.0 |
| Travelling Salesman | **100.0** | **100.0** | 28.8 | 33.6 | 0.0 |

Table 3: The success rates (%) after $N$ node expansions in Sliding Tile Puzzle computed only on problem instances *solvable by HIPS*. $\langle N \rangle$ is the mean number of expansions needed to find a solution.

| $N$ | 50 | 75 | 100 | 200 | 500 | $\langle N \rangle$ |
|---|---|---|---|---|---|---|
| HIPS | 9.2 | 38.2 | 59.8 | 90.8 | 99.7 | 108.9 |
| HIPS-$\varepsilon$, $\varepsilon = 10^{-5}$ | **17.3** | **48.4** | **68.2** | **93.5** | **99.9** | **95.6** |

in the Travelling Salesman, where the results are approximately equal. Next, we analyze whether the increased number of solved puzzles is only because of the high-level search failing to find a solution to some puzzles. In Table 3, we consider the Sliding Tile Puzzle problem and report the same results as in Table 1 considering only the puzzles *solvable by HIPS*. The results suggest that the complete search can speed up finding a correct solution.

In Fig. 3, we analyze the effect of the hyperparameter $\varepsilon$ on the number of solved puzzles. We report the ratio of the number of unsolved puzzles by HIPS-$\varepsilon$ to the number of unsolved puzzles by HIPS as a function of $N$, the number of node expansions. We observe that HIPS-$\varepsilon$ outperforms HIPS in all environments except TSP, and extensive hyperparameter tuning to select a good value of $\varepsilon$ is rarely necessary. The plots also support our hypothesis about a small value of $\varepsilon$ being often better despite the worse worst-case performance, as $\varepsilon = 10^{-3}$ is often superior to $\varepsilon = 0.1$. The augmented search strategy suggested by Zawalski et al. [45] (which we denote as $\varepsilon \to 0$) is sometimes but not always the best strategy. We analyze the impact of the value of $\varepsilon$ in Appendix I.

**Generalization to Out-of-Distribution Puzzles**    In the following experiment, we demonstrate one of the most attractive properties of complete search: its ability to enhance the high-level search policy to transfer to new (potentially more complex) search problems. We evaluate HIPS-$\varepsilon$ on Box-World in which the following changes have been made in comparison to the trajectories in $\mathcal{D}$: the number of distractor keys has been increased from three to four, and the colors of distractors changed such that there are longer distractor trajectories that lead into dead-end states, making it more difficult to detect distractor keys. After these changes to the environment, the share of puzzles HIPS can solve drops from over 99% to approximately 66% (see Fig. 4). However, the 100% solution rate can be retained by performing complete search. Note here that higher values of $\varepsilon$ seem superior to lower values when the generator is worse, which is aligned with our analysis in Section 3.4.

**Role of the Heuristic Factor**    Orseau and Lelis [27] observed that for pure low-level search, using a learned heuristic in addition to a search policy $\pi$ can lead to significantly improved search performance. We evaluated the performance of HIPS-$\varepsilon$ with different heuristic factors and corresponding PHS evaluation functions. We compared the heuristic factor and evaluation function (4) proposed by us, the Levin-TS inspired evaluation function $\varphi_{\text{LevinTS}} = g(n)/\pi(n)$ that does not use a heuristic, and

Table 4: The success rates (%) of HIPS-$\varepsilon$ with different PHS evaluation functions. For all environments, we use $\varepsilon = 10^{-3}$. Search without heuristic fails in TSP due to running out of memory.

|  | *Sokoban* | | | *Sliding Tile Puzzle* | | | *Box-World* | | *TSP* | | |
|---|---|---|---|---|---|---|---|---|---|---|---|
| $N$ | 50 | 100 | 200 | 50 | 100 | 200 | 5 | 10 | 20 | 50 | 100 |
| $\hat{\varphi}_{\hat{h}}$ (ours) | **82.5** | **88.8** | **92.9** | **16.3** | **68.6** | **93.7** | **89.0** | **99.1** | **18.3** | **82.4** | **96.1** |
| $\varphi_{\text{LevinTS}}$ | 66.4 | 80.1 | 88.8 | 0.0 | 0.0 | 0.5 | 27.4 | 66.3 | N/A | N/A | N/A |
| $\varphi_{\text{depth}}$ | 71.8 | 83.6 | 91.5 | 0.0 | 0.2 | 1.2 | 37.3 | 75.8 | 0.0 | 0.0 | 0.0 |
| $\varphi_{\text{dist}}$ | 68.6 | 81.3 | 88.7 | 0.0 | 0.1 | 0.7 | 30.4 | 70.0 | 0.0 | 0.0 | 0.0 |

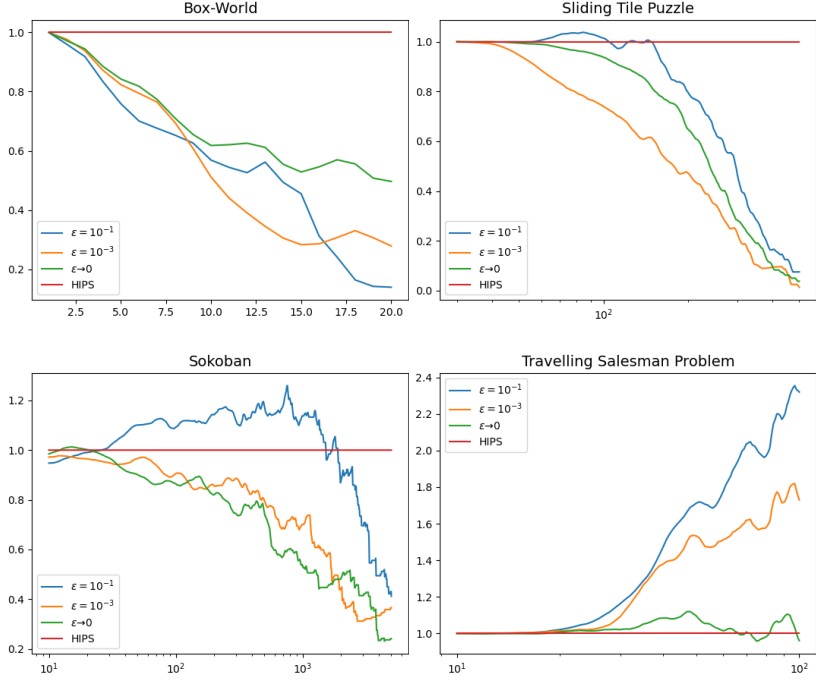

Figure 3: The ratio of the number of unsolved puzzles to the number of unsolved puzzles by HIPS as a function of the number of node expansions $N$ (x-axis). Values below 1 indicate that the complete search is superior to the high-level search. HIPS-$\varepsilon$ outperforms HIPS in every environment except TSP, where high-level actions are sufficient for solving every problem instance.

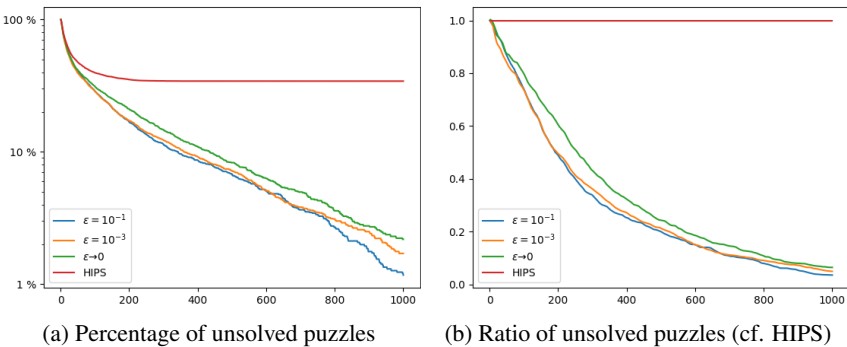

(a) Percentage of unsolved puzzles        (b) Ratio of unsolved puzzles (cf. HIPS)

Figure 4: The percentage of puzzles remaining unsolved (y-axis) depending on the number of node expansions (x-axis) for complete search with different values of $\varepsilon$ and only high-level search (left), and the ratio of unsolved puzzles in comparison with HIPS depending on the number of node expansions (right), when the methods have been evaluated on an out-of-distribution variant of Box-World.

naive A*-inspired evaluation functions [9] which correspond to the variant $\mathrm{PHS}_h$ in [27]:

$$\varphi_{\text{depth}} = (g(n) + h(n))/\pi(n), \qquad \varphi_{\text{dist}} = (l(n) + h(n))/\pi(n).$$

The results in Table 4 show that a heuristic function is crucial for reaching a competitive search performance in most environments. In particular, on TSP, where it was observed that training a good VQVAE prior (which we use as $\pi_{\text{SG}}$) is very difficult [18], the search heavily relies on the learned heuristic, and the search fails due to running out of memory. Furthermore, naively using A*-inspired evaluation functions fails to be competitive.

# 5 Conclusions

Subgoal search algorithms can effectively address complex reasoning problems that require long-term planning. However, these algorithms may fail to find a solution to solvable problems. We have presented and analyzed HIPS-$\varepsilon$, an extension to a recently proposed subgoal search algorithm HIPS. We achieve this by augmenting the subgoal-level search with low-level actions. As a result, HIPS-$\varepsilon$ is guaranteed to discover a solution if a solution exists and it has access to an accurate environment model. HIPS-$\varepsilon$ outperforms HIPS and other baseline methods in terms of search loss and solution rate. Furthermore, the results demonstrate that augmenting the search with low-level actions can improve the planning performance even if the subgoal search could solve the puzzle without them, and the proposed algorithm is not hyperparameter-sensitive. HIPS-$\varepsilon$ enables using subgoal search in discrete settings where search completeness is critical.

In the future, we would like to tackle some of the limitations of our work. Our search paradigm could be applied to other subgoal search methods than HIPS and search algorithms than PHS. Combining HIPS-$\varepsilon$ with recursive search methods that split the problem into smaller segments could enable scaling to even longer problem horizons. Our analysis of the complete subgoal search assumes deterministic environments and access to transition dynamics. HIPS-$\varepsilon$ can also function without access to the transition dynamics, but the completeness guarantee is lost. Many real-world problems are partially or fully continuous. There are also problems with infinite discrete action spaces that HIPS-$\varepsilon$ is not particularly suited for, and we assume discrete state representations. HIPS-$\varepsilon$ also assumes that a solution exists. Modifying HIPS-$\varepsilon$ for the continuous setting could enable solving real-world robotics tasks. HIPS-$\varepsilon$ also showed potential at transfer learning by efficiently solving more complex Box-World tasks than those seen during training. Applying curriculum learning by learning the initial model offline and then learning to progressively solve harder problem instances with online fine-tuning is a promising direction for future research.

## Acknowledgments and Disclosure of Funding

We acknowledge the computational resources provided by the Aalto Science-IT project and CSC, Finnish IT Center for Science. The work was funded by Research Council of Finland (aka Academy of Finland) within the Flagship Programme, Finnish Center for Artificial Intelligence (FCAI). J. Pajarinen was partly supported by Research Council of Finland (aka Academy of Finland) (345521).

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
