## A    Potential Negative Societal Impacts

We have not trained our models with sensitive or private data, and we emphasize that our model's direct applicability to real-world decision-making concerning humans is currently limited. Nevertheless, we must remain vigilant about potential unintended uses that could have harmful implications, particularly in contexts like military or law enforcement applications or other scenarios that are difficult to anticipate. While we acknowledge that there may not be immediate negative societal impacts associated with our work on complete search and HIPS-$\varepsilon$, it is also essential to consider the potential long-term consequences. For instance, when applying HIPS-$\varepsilon$ to data containing human data or collected with humans, the issues of fairness, privacy, and sensitivity must be taken into account.

## B    Formal Problem Definition

We adopt the formal problem definition from Orseau and Lelis [27]. Assume that there is a single-agent task $k$ that is modeled as a fully observable Markov Decision Process $\mathcal{M} = (\mathcal{S}, \mathcal{A}, P, R, \gamma, \mathcal{S}_0)$, where $\mathcal{S}$ is the state space, $\mathcal{A}$ the action space, $P$ the transition dynamics, $R$ the reward function, $\gamma$ the discount factor, and $\mathcal{S}_0$ the set of initial states. We also assume that the state space is discrete, the action space is discrete and finite, and the transition function is deterministic. The reward is equal to one when a terminal state has been reached and zero otherwise, that is, the tasks are goal-oriented.

We assume that there is a search algorithm $S$, and an associated task search loss $L_k$. The objective is to find a solution to the task $k$ by performing tree search using the algorithm $S$. Let $\mathcal{N}$ be the set of all possible nodes in the tree, and let $\mathcal{N}_{\mathcal{G}}$ be the set of solution nodes. Our objective is to minimize the search loss $\min_{n^* \in \mathcal{N}_{\mathcal{G}}} L_k(S, n^*)$. We assume that every time the search expands a node, we incur a node loss $L(n)$, where $L(n) : \mathcal{N} \to [0, \infty]$ is a loss function defined for all nodes $n \in \mathcal{N}$, and the task search loss for node $n$, $L_k(S, n)$ is equivalent to the sum of individual losses $L(n')$ for all nodes $n'$ that have been expanded before expanding node $n$.

During learning, we adopt the imitation learning (offline) setting: the agent must learn to solve the task and minimize the search loss without interacting with the environment. Instead, there is a dataset $\mathcal{D}$ of trajectories $\tau = \{s_0, a_0, s_1, \ldots, a_{T-1}, s_T\}$. The trajectories in the dataset are goal-reaching but can be highly suboptimal, that is, the expert does not reach the terminal state in the fastest way possible.

## C    Derivation of Complete Search Heuristic

Given a node $n$, its set of descendants $\text{desc}_*(n)$ (all possible nodes following after $n$), and the set of goal nodes $\mathcal{N}_{\mathcal{G}}$, Orseau and Lelis [27] argue that the ideal heuristic factor is equal to

$$\eta(n) = \min_{n^* \in \text{desc}_*(n) \cap \mathcal{N}_{\mathcal{G}}} \frac{g(n^*)/\pi(n^*)}{g(n)/\pi(n)}. \tag{8}$$

Then, $g(n^*)$ can be approximated as the sum of the path loss $g(n)$ to the current node $n$, and the predicted additional loss from the current node $n$ to the closest descendant target node $n^*$. Assuming a path from the root node $n_0$ to $n$, $(n_0, n_1, \ldots, n)$, and that the algorithm incurs a loss of $L(n_i)$ for expanding any node $n_i$, the path loss $g(n)$ of $n$ is $\sum_{n_i \in (n_0, \ldots, n)} L(n_i)$, and the additional loss from the node $n$ to the target $n^*$ should be equal to $\sum_{n_i \in (n', \ldots, n^*)} L(n_i)$, where $n'$ is the child node of $n$ on the path to $n^*$. Then, $g(n^*) = g(n) + \sum_{n_i \in (n', \ldots, n^*)} L(n_i)$.

We know the value of $g(n)$ but we do not know the path $(n, n', \ldots, n^*)$ before executing the search, so we need to approximate $\sum_{n_i \in (n', \ldots, n^*)} L(n_i)$ using the learned heuristic $h(n)$ trained to predict the number of low-level steps from $n$ to $n^*$ [27]. However, $L(n_i)$ may not directly depend on the number of low-level steps between $n_i$ and its parent. In particular, we have defined that for HIPS-$\varepsilon$, $\forall n' \in \mathcal{N} : L(n') = 1$, where $\mathcal{N}$ is the complete search space. Therefore, we would need to know the predicted length of the path $(n, \ldots, n^*)$ in terms of the number of nodes, not low-level actions, which is what $h(n)$ predicts. To circumvent this, we assume that the average number of low-level actions between nodes is approximately constant, and we propose scaling the heuristic $h(n)$ by dividing it by the average number of low-level actions per node on the path $(n_0, \ldots, n)$. This results in the scaled

heuristic $\hat{h}(n) = h(n)/\frac{l(n)}{g(n)} = \frac{h(n)g(n)}{l(n)}$, where $l(n)$ is the number of low-level actions on the path from $n_0$ to $n$. Hence, the approximation of $g(n^*)$ becomes

$$g(n^*) = g(n) + \hat{h}(n) = g(n) + \frac{h(n)g(n)}{l(n)} = g(n)\left(1 + \frac{h(n)}{l(n)}\right). \tag{9}$$

Note that this scaled heuristic and the corresponding approximation for $g(n^*)$ are valid for losses $L(n)$ other than the constant one as long as $g(n)$ and $l(n)$ are positively correlated.

Orseau and Lelis [27] proposed to approximate the probability $\pi(n^*)$ as $\pi(n^*) = [\pi(n)^{1/g(n)}]^{g(n)+h(n)}$. This can be interpreted as first taking the average conditional probability $p = [\pi(n)^{1/g(n)}]$ along the path from the root to $n$, and then scaling it to the full length $g(n) + h(n)$ as $p^{g(n)+h(n)}$. In our case, $h(n)$ estimates the distance to the terminal node in terms of low-level actions. Therefore, we use our scaled heuristic $\hat{h}(n)$ to get the estimate in terms of search nodes instead. This leads to a new approximation for $\pi(n^*)$:

$$\pi(n^*) = [\pi(n)^{1/g(n)}]^{g(n)+\hat{h}(n)} \tag{10}$$

$$= [\pi(n)^{1/g(n)}]^{g(n)+\frac{h(n)g(n)}{l(n)}} \tag{11}$$

$$= \pi(n)^{1+\frac{h(n)}{l(n)}} \tag{12}$$

Then, we insert the approximations for $g(n^*)$ and $\pi(n^*)$ into (8), with $h(n)$ predicting the distance to the closest terminal node and thus allowing us to drop the min, similarly as Orseau and Lelis [27].

$$\eta(n) = \min_{n^*\in\text{desc}_*(n)\cap\mathcal{N}_\mathcal{G}} \frac{g(n^*)/\pi(n^*)}{g(n)/\pi(n)} \tag{13}$$

$$= \min_{n^*\in\text{desc}_*(n)\cap\mathcal{N}_\mathcal{G}} \frac{g(n^*)}{g(n)} \cdot \frac{\pi(n)}{\pi(n^*)} \tag{14}$$

$$= \left(1 + \frac{h(n)}{l(n)}\right) \cdot \frac{\pi(n)}{\pi(n)^{1+\frac{h(n)}{l(n)}}} \tag{15}$$

$$= \frac{1 + h(n)/l(n)}{\pi^{h(n)/l(n)}}, \tag{16}$$

which we denote by $\hat{\eta}_{\hat{h}}$. Finally, we insert (16) into $\varphi(n) = \eta(n)\frac{g(n)}{\pi(n)}$ [27] yielding

$$\varphi(n) = \frac{g(n) \cdot (1 + h(n)/l(n))}{\pi(n)^{1+h(n)/l(n)}}, \tag{17}$$

which we denote by $\hat{\varphi}_{\hat{h}}(n)$ and use as the evaluation function of HIPS-$\varepsilon$.

## D  Experiment Details

The code for HIPS-$\varepsilon$ can be found on GitHub[2]. We implemented the BC policy $\pi_{\text{BC}}(a|s_{j-1})$ by adding a new head to the conditional VQVAE prior $p(e|s)$, which acts as the high-level policy $\pi_{\text{SG}}$ and that has been implemented as a ResNet-based CNN. We used the hyperparameters and networks from [18] for the other components (also shown in Tables 5 and 6). The new hyperparameter of HIPS-$\varepsilon$ is $\varepsilon$. For all four environments, we evaluated the values $10^{-1}, 10^{-3}, 10^{-5}$, and $\varepsilon \to 0$, and chose the best-performing one to be included in the results. For Sokoban and TSP that is $\varepsilon \to 0$, for Box-World it is $\varepsilon = 10^{-3}$, and for Sliding Tile Puzzle $\varepsilon = 10^{-5}$ unless specified otherwise (see also Table 9). The results for the baselines AdaSubS, kSubS, BC, CQL, DT, and HIPS with learned models were copied from [18].

We used the Sokoban implementation in Gym-Sokoban (MIT License) [37] and the Sliding Tile Puzzle implementation from Orseau and Lelis [27] (Apache License 2.0). We ran our experiments using the demonstration dataset from [18]. The dataset contains 10340 trajectories in Sokoban, 5100 in Sliding Tile Puzzle, and 22,100 in Bow-World. The number of trajectories in TSP is unlimited,

---

[2]https://github.com/kallekku/HIPS

but they are of low quality. The datasets also contain a validation set, which we use to evaluate early stopping.

The total number of GPU hours used on this work was approximately 7,500. Approximately 15,000 hours of GPU time in total were used for exploratory experiments during the project. We used an HPC cluster with AMD MI250X GPUs for running the experiments in this paper. Each job was run using a single GPU. We used 6 CPU workers (AMD Trento) per GPU.

Table 5: General hyperparameters of our method.

| Parameter | Value |
|---|---|
| Learning rate for dynamics | $2 \cdot 10^{-4}$ |
| Learning rate for $\pi$, $d$, $V$ | $1 \cdot 10^{-3}$ |
| Learning rate for VQVAE | $2 \cdot 10^{-4}$ |
| Discount rate for REINFORCE | 0.99 |

Table 6: Environment-specific hyperparameters of our method.

| Parameter | Explanation | Sokoban | STP | Box-World | TSP |
|---|---|---|---|---|---|
| $\alpha$ | Subgoal penalty | 0.1 | 0.1 | 0.1 | 0.05 |
| $\beta$ | Beta for VQVAE | 0.1 | 0.1 | 0.1 | 0 |
| $D$ | Codebook dimensionality | 128 | 128 | 128 | 64 |
| $H$ | Subgoal horizon | 10 | 10 | 10 | 50 |
| $K$ | VQVAE codebook size | 64 | 64 | 64 | 32 |
| $(N, D)$ | DRC size | – | – | (3, 3) | – |

# E   Attaining Completeness with Learned Dynamic Models

In the main text, we assumed that we either have access to the environment dynamics and the number of environment steps is not a cost to be minimized or that we have no access to the environment and use a learned model. In the latter case, completeness cannot be guaranteed. However, we can assume a setting where we have access to the environment dynamics, but each step is costly. Hence, the objective is to minimize the number of environment interactions while retaining completeness. To achieve this, we learn a dynamics model and perform a search with it. If a solution is found, we validate it with the environment dynamics. Furthermore, we simulate the consequences of the low-level actions with the known environment dynamics, which allows us to guarantee search completeness and minimize the number of environment steps and search node expansions required to find the solution. The results for this modification can be found in Appendix K.

# F   Results for STP with GBFS and TSP with A*

In the main text, we evaluated HIPS-$\varepsilon$ with PHS* as the underlying search algorithm on all tasks and compared those numbers to the results of HIPS with PHS* as the search algorithm. However, HIPS originally used Greedy Best-First Search (GBFS) in STP and A* in TSP [18]. We evaluated HIPS-$\varepsilon$ with these search algorithms on the environments. We tried two strategies: using low-level actions and subgoals equally and only using low-level actions when the high-level search was about to fail. The latter outperformed the former strategy. In TSP, high-level actions alone are sufficient for solving the task, and in STP, the greedy approach with equal use of the low-level actions suffers from the noisiness of the value function. The results are given in Tables 7 and 8. Using high-level actions improves the signal-to-noise ratio [4], which was the key to good performance.

Table 7: The success rates (%) after performing $N$ node expansions on Sliding Tile Puzzle with GBFS as the underlying search algorithm for each method. The uncertainty is the standard error of the mean. As we cannot control the usage of the low-level actions with $\varepsilon$, we expand low-level actions only when necessary ($\epsilon \to 0$). HIPS corresponds to HIPS-env in [18].

| | Sliding Tile Puzzle (GBFS) | | | |
|---|---|---|---|---|
| $N$ | 50 | 100 | 200 | $\infty$ |
| HIPS | $78.6 \pm 2.6$ | $90.2 \pm 1.8$ | $91.4 \pm 1.6$ | $94.5 \pm 1.0$ |
| AdaSubS | $0.0 \pm 0.0$ | $0.0 \pm 0.0$ | $0.0 \pm 0.0$ | $0.0 \pm 0.0$ |
| kSubS | $0.7 \pm 0.2$ | $79.9 \pm 3.1$ | $89.8 \pm 1.5$ | $93.3 \pm 0.8$ |
| HIPS-$\varepsilon$ | $\mathbf{83.6} \pm 2.1$ | $\mathbf{94.6} \pm 1.3$ | $\mathbf{95.8} \pm 1.2$ | $\mathbf{100.0} \pm 0.0$ |

Table 8: The success rates (%) after performing $N$ node expansions for HIPS and HIPS-$\varepsilon$ on TSP with A* as the underlying search algorithm. The uncertainty is the standard error of the mean. As we cannot control the usage of the low-level actions with $\varepsilon$, we expand the low-level actions only when necessary ($\epsilon \to 0$). HIPS corresponds to HIPS-env in [18].

| | Travelling Salesman Problem (A*) | | | |
|---|---|---|---|---|
| $N$ | 20 | 50 | 100 | $\infty$ |
| HIPS | $\mathbf{54.3} \pm 13.7$ | $\mathbf{99.9} \pm 0.1$ | $\mathbf{100.0} \pm 0.0$ | $\mathbf{100.0} \pm 0.0$ |
| HIPS-$\varepsilon$ | $52.4 \pm 13.3$ | $99.8 \pm 0.2$ | $\mathbf{100.0} \pm 0.0$ | $\mathbf{100.0} \pm 0.0$ |

# G   Full Results with Confidence Intervals

Table 9: The mean success rates (%) after performing $N$ node expansions for different subgoal search algorithms with access to environment dynamics and the standard error of the mean as the uncertainty metric. For HIPS-$\varepsilon$, we use the value of $\varepsilon$ that yields in the best performance: $\varepsilon \to 0$ for Sokoban, $\varepsilon = 10^{-5}$ for Sliding Tile Puzzle, $\varepsilon = 10^{-3}$ for Box-World, and $\varepsilon \to 0$ for Travelling Salesman Problem. HIPS corresponds to HIPS-env in [18] and uses PHS* as the search algorithm in all environments.

| | *Sokoban* | | | |
| --- | --- | --- | --- | --- |
| $N$ | 50 | 100 | 200 | $\infty$ |
| PHS* (low-level search) | $0.2 \pm 0.1$ | $2.4 \pm 0.4$ | $16.2 \pm 1.3$ | $\mathbf{100} \pm 0.0$ |
| HIPS (high-level search) | $82.0 \pm 0.7$ | $87.8 \pm 0.5$ | $91.6 \pm 0.4$ | $97.9 \pm 0.4$ |
| AdaSubS (high-level search) | $76.4 \pm 0.5$ | $82.2 \pm 0.5$ | $85.7 \pm 0.6$ | $91.3 \pm 0.5$ |
| kSubS (high-level search) | $69.1 \pm 2.2$ | $73.1 \pm 2.2$ | $76.3 \pm 1.9$ | $90.5 \pm 1.0$ |
| HIPS-$\varepsilon$ (complete search) | $\mathbf{84.3} \pm 1.1$ | $\mathbf{89.5} \pm 1.1$ | $\mathbf{93.1} \pm 0.6$ | $\mathbf{100} \pm 0.0$ |

| | *Sliding Tile Puzzle* | | | |
| --- | --- | --- | --- | --- |
| $N$ | 50 | 100 | 200 | $\infty$ |
| PHS* (low-level search) | $0.0 \pm 0.0$ | $0.0 \pm 0.0$ | $0.0 \pm 0.0$ | $\mathbf{100} \pm 0.0$ |
| HIPS (high-level search) | $8.7 \pm 1.2$ | $56.8 \pm 4.5$ | $86.3 \pm 2.1$ | $95.0 \pm 0.8$ |
| AdaSubS (high-level search) | $0.0 \pm 0.0$ | $0.0 \pm 0.0$ | $0.0 \pm 0.0$ | $0.0 \pm 0.0$ |
| kSubS (high-level search) | $0.7 \pm 0.2$ | $\mathbf{79.9} \pm 3.1$ | $89.8 \pm 1.5$ | $93.3 \pm 0.8$ |
| HIPS-$\varepsilon$ (complete search) | $\mathbf{18.5} \pm 1.9$ | $69.5 \pm 3.9$ | $\mathbf{93.8} \pm 1.7$ | $\mathbf{100} \pm 0.0$ |

| | *Box-World* | | | |
| --- | --- | --- | --- | --- |
| $N$ | 5 | 10 | 30 | $\infty$ |
| PHS* (low-level search) | $0.0 \pm 0.0$ | $0.1 \pm 0.1$ | $2.2 \pm 0.5$ | $\mathbf{100} \pm 0.0$ |
| HIPS (high-level search) | $86.3 \pm 0.7$ | $97.9 \pm 0.3$ | $99.9 \pm 0.0$ | $99.9 \pm 0.0$ |
| HIPS-$\varepsilon$ (complete search) | $\mathbf{89.7} \pm 0.6$ | $\mathbf{98.9} \pm 0.2$ | $\mathbf{100} \pm 0.0$ | $\mathbf{100} \pm 0.0$ |

| | *Travelling Salesman Problem* | | | |
| --- | --- | --- | --- | --- |
| $N$ | 20 | 50 | 100 | $\infty$ |
| PHS* (low-level search) | $0.0 \pm 0.0$ | $0.0 \pm 0.0$ | $0.0 \pm 0.0$ | $\mathbf{100} \pm 0.0$ |
| HIPS (high-level search) | $\mathbf{19.6} \pm 6.0$ | $\mathbf{88.1} \pm 4.1$ | $97.7 \pm 1.1$ | $\mathbf{100} \pm 0.0$ |
| AdaSubS (high-level search) | $0.0 \pm 0.0$ | $0.0 \pm 0.0$ | $0.6 \pm 0.3$ | $21.2 \pm 0.9$ |
| kSubS (high-level search) | $0.0 \pm 0.0$ | $1.5 \pm 0.6$ | $40.4 \pm 9.1$ | $87.9 \pm 3.1$ |
| HIPS-$\varepsilon$ (complete search) | $17.9 \pm 5.8$ | $87.4 \pm 4.1$ | $\mathbf{97.9} \pm 1.2$ | $\mathbf{100} \pm 0.0$ |

Table 10: The success rates (%) of different algorithms without access to environment dynamics in Sokoban, Sliding Tile Puzzle, and TSP with the standard errors of the mean as the uncertainty metric. HIPS-$\varepsilon$ outperforms the baselines and can solve 100% of the puzzles when the environment dynamics are easy to learn, but when they are more difficult, occasional failures cannot be excluded.

|  | HIPS | HIPS-$\varepsilon$ | BC | CQL | DT |
|---|---|---|---|---|---|
| Sokoban | $97.5 \pm 0.6$ | $\mathbf{100.0 \pm 0.0}$ | $18.7 \pm 0.7$ | $3.3 \pm 0.4$ | $36.7 \pm 1.2$ |
| Sliding Tile Puzzle | $94.7 \pm 1.0$ | $\mathbf{100.0 \pm 0.0}$ | $82.5 \pm 2.2$ | $11.7 \pm 3.3$ | $0.0 \pm 0.0$ |
| Travelling Salesman | $99.9 \pm 0.1$ | $\mathbf{100 \pm 0.0}$ | $28.8 \pm 8.5$ | $33.6 \pm 2.6$ | $0.0 \pm 0.0$ |

Table 11: The success rates (%) after $N$ node expansions in Sliding Tile Puzzle computed only on problem instances *solvable by HIPS*. $\langle N \rangle$ is the mean number of expansions needed to find a solution. The uncertainty metric is the standard error of the mean. We used $\varepsilon = 10^{-5}$.

| $N$ | 50 | 75 | 100 | 200 | 500 | $\langle N \rangle$ |
|---|---|---|---|---|---|---|
| HIPS | $9.2 \pm 1.3$ | $38.2 \pm 3.6$ | $59.8 \pm 4.7$ | $90.8 \pm 2.1$ | $99.7 \pm 0.2$ | $108.9 \pm 6.9$ |
| HIPS-$\varepsilon$ | $\mathbf{17.3 \pm 1.6}$ | $\mathbf{48.4 \pm 4.3}$ | $\mathbf{68.2 \pm 4.0}$ | $\mathbf{93.5 \pm 1.7}$ | $\mathbf{99.9 \pm 0.1}$ | $\mathbf{95.6 \pm 5.7}$ |

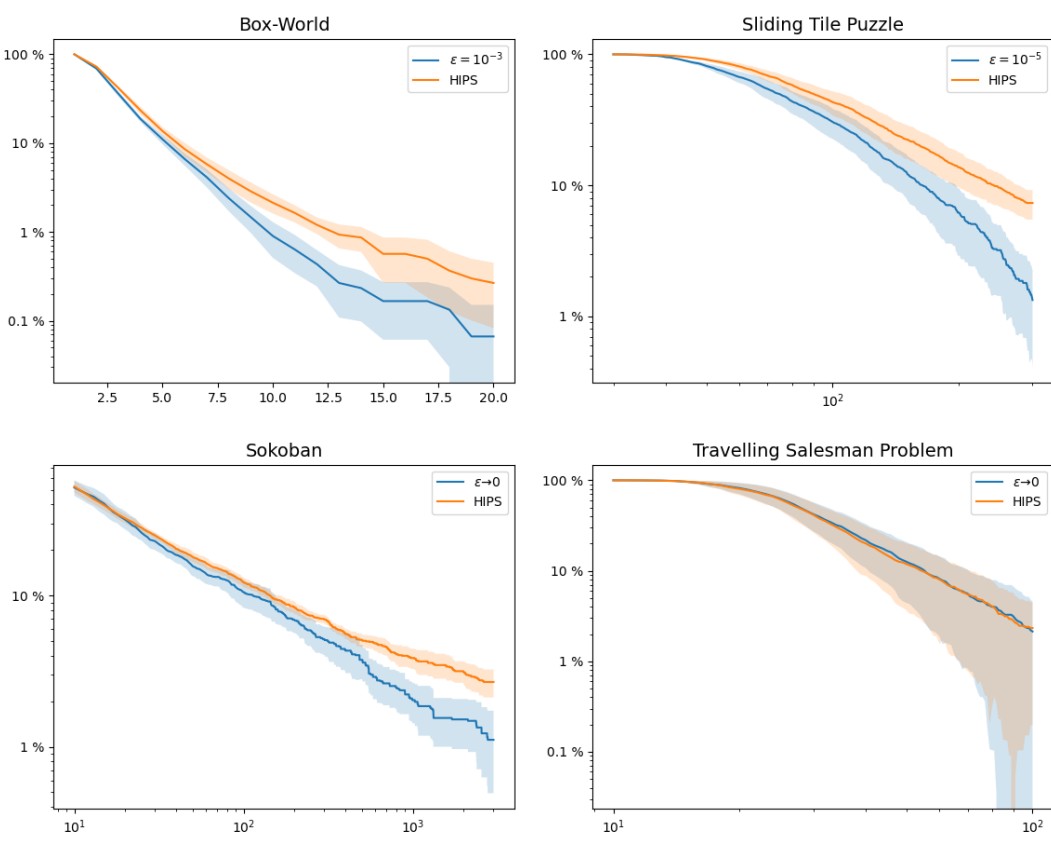

Figure 5: The percentage of puzzles remaining unsolved (y-axis) depending on the number of node expansions (x-axis) for complete search with the best values of $\varepsilon$. The shaded area is two standard errors (see also Table 9). The differences between HIPS-$\varepsilon$ and HIPS are statistically significant except for TSP, where $\epsilon \to 0$ and HIPS are equal since the low-level actions are never used in practice.

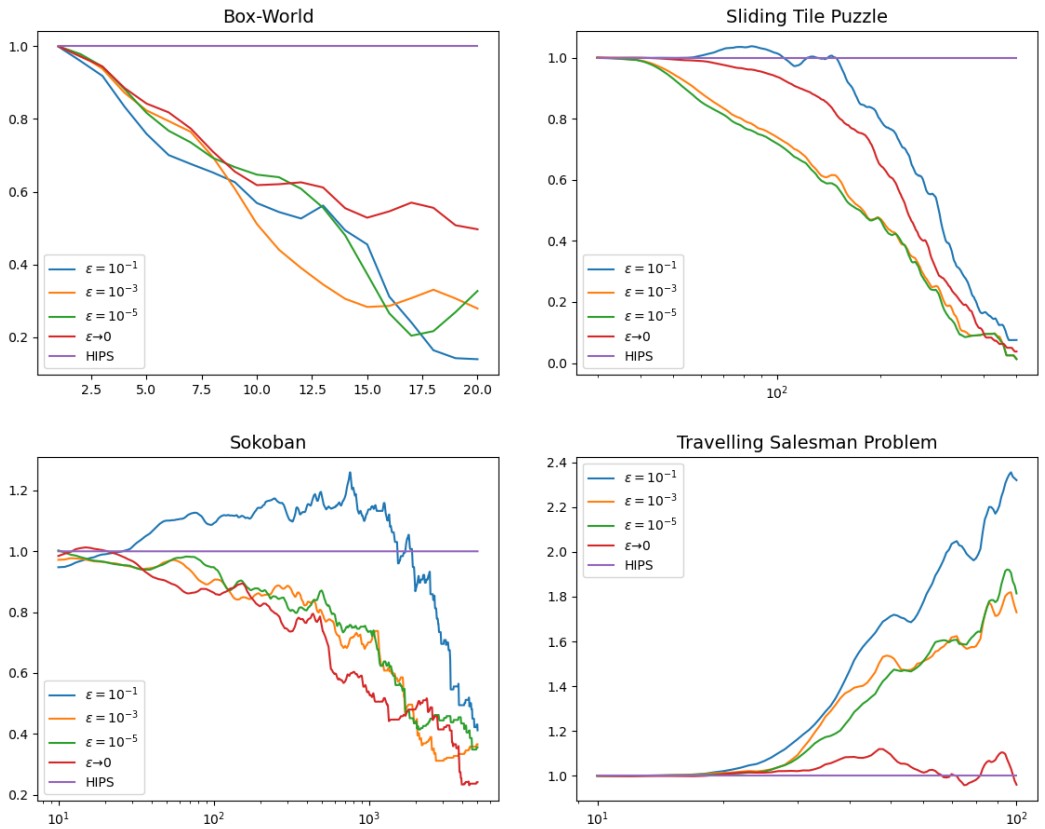

Figure 6: The ratio of the number of unsolved puzzles to the number of unsolved puzzles by HIPS as a function of the number of node expansions $N$ (x-axis) with multiple values of $\varepsilon$. Values below 1 indicate that the complete search is superior to the high-level search. HIPS-$\varepsilon$ outperforms HIPS in every environment except TSP, where high-level actions are sufficient for solving every problem instance.

Table 12: The mean success rates (%) after performing $N$ node expansions for low-level PHS*, high-level HIPS, and HIPS-$\varepsilon$ with access to environment dynamics and the standard error of the mean as the uncertainty metric. The algorithms have been evaluated on a more difficult variant of Box-World without any tuning to the new environment. HIPS corresponds to HIPS-env in [18].

| | Box-World (OoD) | | | | |
|---|---|---|---|---|---|
| $N$ | 10 | 50 | 200 | 1000 | $\infty$ |
| PHS* | $0.5 \pm 0.2$ | $5.0 \pm 0.8$ | $28.6 \pm 1.1$ | $31.4 \pm 1.0$ | $\mathbf{100.0} \pm 0.0$ |
| HIPS | $12.3 \pm 1.5$ | $52.3 \pm 3.1$ | $64.8 \pm 2.5$ | $65.8 \pm 2.4$ | $65.8 \pm 2.4$ |
| HIPS-$\varepsilon$, $\varepsilon = 10^{-1}$ | $27.0 \pm 2.4$ | $59.2 \pm 2.5$ | $\mathbf{83.1} \pm 1.4$ | $\mathbf{98.3} \pm 0.3$ | $\mathbf{100.0} \pm 0.0$ |
| HIPS-$\varepsilon$, $\varepsilon = 10^{-3}$ | $\mathbf{27.3} \pm 2.7$ | $\mathbf{60.5} \pm 3.2$ | $82.8 \pm 1.7$ | $\mathbf{98.3} \pm 0.4$ | $\mathbf{100.0} \pm 0.0$ |
| HIPS-$\varepsilon$, $\varepsilon = 10^{-5}$ | $\mathbf{27.3} \pm 2.3$ | $59.7 \pm 2.7$ | $80.7 \pm 1.6$ | $98.0 \pm 0.4$ | $\mathbf{100.0} \pm 0.0$ |
| HIPS-$\varepsilon$, $\varepsilon \rightarrow 0$ | $26.2 \pm 2.1$ | $58.3 \pm 2.8$ | $78.9 \pm 2.0$ | $97.8 \pm 0.4$ | $\mathbf{100.0} \pm 0.0$ |

Table 13: The success rates (%) of HIPS-$\varepsilon$ with different PHS evaluation functions including the standard errors of the mean as the uncertainty metric. For all environments, we use $\varepsilon = 10^{-3}$. Search without heuristic fails on TSP due to running out of memory.

| | Sokoban | | | Sliding Tile Puzzle | | |
|---|---|---|---|---|---|---|
| $N$ | 50 | 100 | 200 | 50 | 100 | 200 |
| $\hat{\varphi}_{\hat{h}}$ (ours) | $\mathbf{82.5} \pm 0.9$ | $\mathbf{88.8} \pm 0.7$ | $\mathbf{92.9} \pm 0.4$ | $\mathbf{16.3} \pm 1.9$ | $\mathbf{68.6} \pm 4.0$ | $\mathbf{93.7} \pm 1.7$ |
| $\varphi_{\text{LevinTS}}$ | $66.4 \pm 2.7$ | $80.1 \pm 2.1$ | $88.8 \pm 1.2$ | $0.0 \pm 0.0$ | $0.0 \pm 0.0$ | $0.5 \pm 0.5$ |
| $\varphi_{\text{depth}}$ | $71.8 \pm 2.3$ | $83.6 \pm 1.5$ | $91.5 \pm 0.8$ | $0.0 \pm 0.0$ | $0.2 \pm 0.1$ | $1.2 \pm 0.4$ |
| $\varphi_{\text{dist}}$ | $68.6 \pm 3.2$ | $81.3 \pm 2.4$ | $88.7 \pm 1.5$ | $0.0 \pm 0.0$ | $0.1 \pm 0.1$ | $0.7 \pm 0.3$ |
| | Box-World | | | Travelling Salesman Problem | | |
| $N$ | 5 | 10 | 20 | 20 | 50 | 100 |
| $\hat{\varphi}_{\hat{h}}$ (ours) | $\mathbf{89.0} \pm 0.7$ | $\mathbf{99.1} \pm 0.2$ | $99.9 \pm 0.0$ | $\mathbf{18.3} \pm 5.5$ | $\mathbf{82.4} \pm 5.5$ | $\mathbf{96.1} \pm 1.5$ |
| $\varphi_{\text{LevinTS}}$ | $27.4 \pm 1.6$ | $66.3 \pm 1.3$ | $94.0 \pm 0.5$ | N/A | N/A | N/A |
| $\varphi_{\text{depth}}$ | $37.3 \pm 2.0$ | $75.8 \pm 1.7$ | $95.9 \pm 0.4$ | $0.0 \pm 0.0$ | $0.0 \pm 0.0$ | $0.0 \pm 0.0$ |
| $\varphi_{\text{dist}}$ | $30.4 \pm 2.2$ | $70.0 \pm 1.8$ | $94.7 \pm 0.5$ | $0.0 \pm 0.0$ | $0.0 \pm 0.0$ | $0.0 \pm 0.0$ |

# H    Out-of-Distribution Evaluation on Sokoban

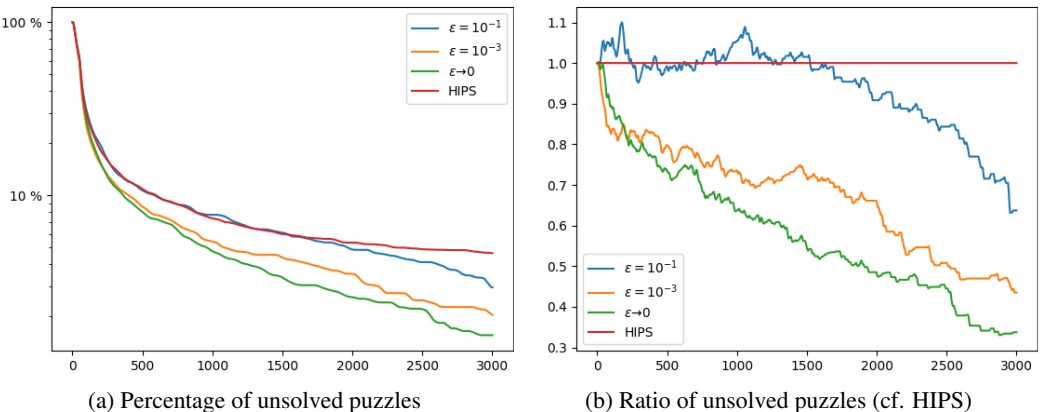

(a) Percentage of unsolved puzzles                    (b) Ratio of unsolved puzzles (cf. HIPS)

Figure 7: Evaluation of HIPS and HIPS-$\varepsilon$ on an out-of-distribution variant of Sokoban with 6 boxes. The percentage of puzzles remaining unsolved (y-axis) depending on the number of node expansions (x-axis) for complete search with different values of $\varepsilon$ and only high-level search (left), and the ratio of unsolved puzzles in comparison with HIPS depending on the number of node expansions (right).

To further analyze how the complete subgoal search proposed by us affects the out-of-distribution generalization abilities of hierarchical planning algorithms, we evaluate HIPS-$\varepsilon$ and HIPS on Sokoban puzzles with six boxes without any adaptation to the models that have been trained on Sokoban with four boxes. The results as a function of search node expansions have been plotted in Figure 7. Our results demonstrate that HIPS-$\varepsilon$ outperforms HIPS, and when the value of $\varepsilon$ is suitably selected, it can solve almost all puzzles within a reasonable number of search node expansions, whereas the performance of HIPS stagnates quite early.

# I   Impact of $\varepsilon$ on Search

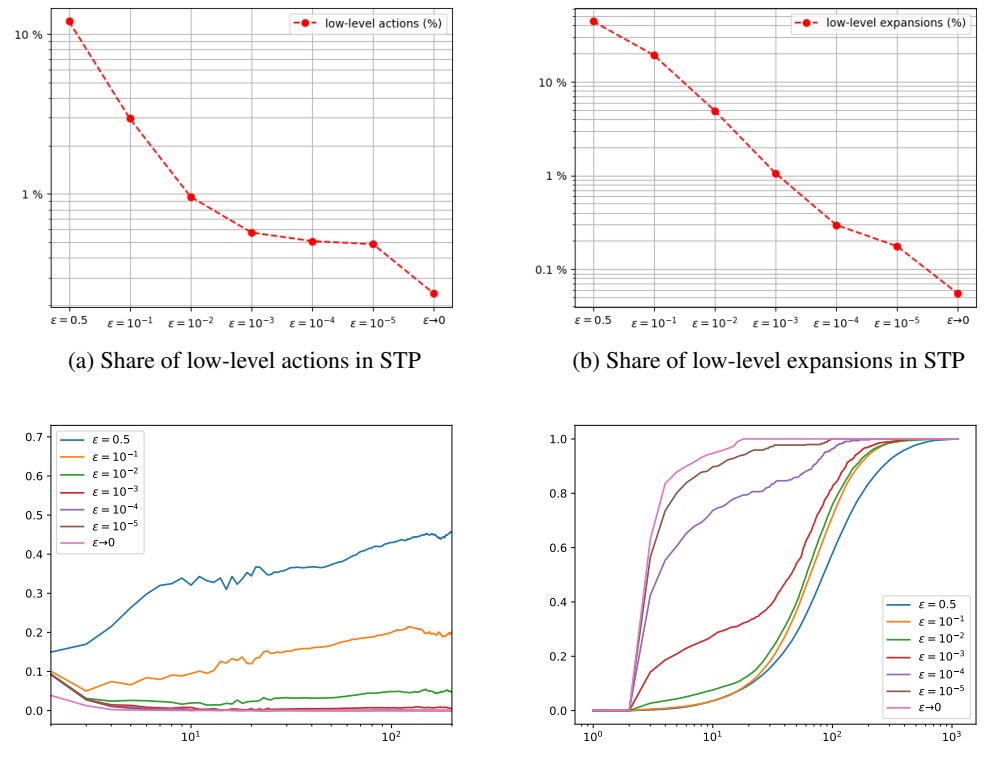

(a) Share of low-level actions in STP

(b) Share of low-level expansions in STP

(c) Share of low-level expansions w.r.t. expansions

(d) Cumulative distribution of low-level expansions

Figure 8: a) The percentage of low-level actions in the solutions found by HIPS-$\varepsilon$ for different values of $\varepsilon$ in Sliding Tile Puzzle. b) The percentage of expansions, where the node corresponds to a low-level action, for different values of $\varepsilon$ in STP. c) The share of low-level expansions in STP with respect to the number of nodes expanded so far in the search. d) The cumulative distribution of low-level expansions in STP for different values of $\varepsilon$.

In Figure 8, we plot how changing the value of $\varepsilon$ impacts the search. In Figure 8a, we plot the value of $\varepsilon$ on the x-axis and the share of low-level actions in the final solutions returned by the search on the y-axis. The greater the value of $\varepsilon$, the larger the share of low-level actions in the solutions found by the search, which is expected behavior. Similarly, Figure 8b shows that when the value of $\varepsilon$ diminishes, fewer search nodes corresponding to low-level actions are expanded. In Figure 8c, we plot the number of search nodes expanded on the x-axis and the probability of low-level search node expansions on the y-axis. We see that for larger values of $\varepsilon$, the relative share of low-level expansions grows as the search progresses, whereas it diminishes for smaller values of $\varepsilon$. Finally, Figure 8d contains the cumulative distribution of low-level expansions as a function of the total number of search node expansions. For instance, when $\varepsilon \to 0$, almost no low-level nodes are expanded after the first ten node expansions.

# J    Wall-Clock Time Evaluations

So far, we have used the number of search node expansions as the evaluation metric for the search efficiency. However, the wall-clock duration of the search is also highly relevant for the search algorithm. Therefore, we compared HIPS-$\varepsilon$ to low-level search PHS* and the high-level search algorithms HIPS, kSubS, and AdaSubS and measured the running times. PHS*, kSubS, and AdaSubS use the environment simulators, and for HIPS and HIPS-$\varepsilon$, we measured the running time with both the environment simulators and learned models.

To make the results of HIPS and HIPS-$\varepsilon$ comparable, we improved HIPS with re-planning if the agent failed to execute the found trajectory due to model incorrectness, which explains why our results for HIPS on Box-World are greatly superior to those in [18]. In some environments, the learned model was significantly faster than the environment simulator, and in other environments, the environment simulator was quicker. The results are shown in Table 14 and illustrated in Figure 9. The results show that HIPS-$\varepsilon$ (ours) outperforms HIPS in Sokoban and STP, and is approximately equal in Box-World and TSP in terms of the running time. kSubS and AdaSubS are not competitive with the HIPS-based algorithms in our experiments, most likely due to the autoregressive generative networks. Low-level search with PHS* is clearly the best in Sliding Tile Puzzle, approximately equal in Box-World, and clearly outperformed by HIPS-$\varepsilon$ in Sokoban and TSP.

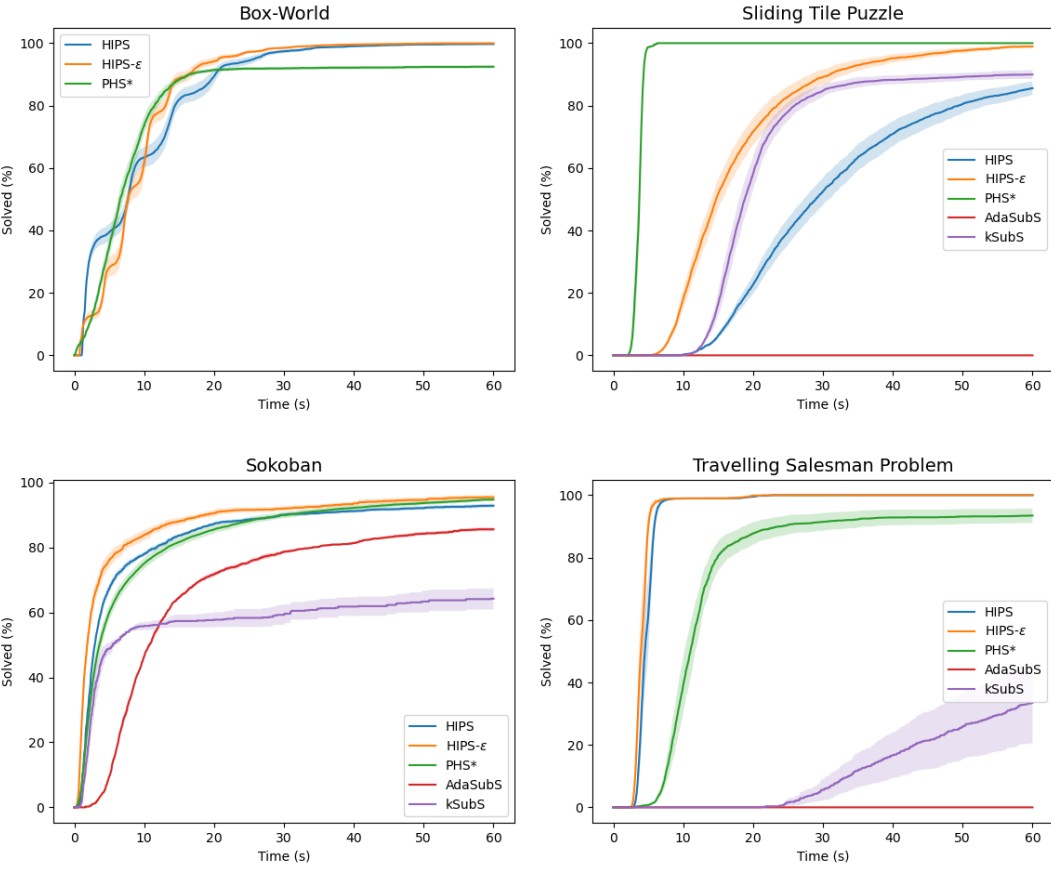

Figure 9: The mean percentage of puzzles solved (%) as a function of the running time in seconds for different search methods. The shaded area is one standard error. For HIPS-$\varepsilon$, we use the same values of $\varepsilon$ as in Table 9. We use PHS* as the underlying search algorithm in Sokoban, STP and Box-World, and A* in TSP.

Table 14: The mean success rates (%) after $s$ seconds of running time for different subgoal search algorithms and the standard error of the mean as the uncertainty metric. For HIPS-$\varepsilon$, we use the same values of $\varepsilon$ as in Table 9. We use PHS* as the underlying search algorithm in Sokoban, STP and Box-World, and A* in TSP.

| | Sokoban | | | | | |
|---|---|---|---|---|---|---|
| $s$ | 1 | 2 | 5 | 10 | 20 | 60 |
| PHS* | $6.1 \pm 1.0$ | $27.3 \pm 3.3$ | $60.0 \pm 2.3$ | $75.1 \pm 1.7$ | $85.6 \pm 1.2$ | $94.8 \pm 0.6$ |
| HIPS | $5.0 \pm 0.6$ | $30.9 \pm 1.8$ | $67.3 \pm 1.2$ | $78.0 \pm 0.8$ | $87.3 \pm 0.5$ | $92.9 \pm 0.6$ |
| HIPS-env | $0.5 \pm 0.2$ | $7.8 \pm 1.2$ | $45.0 \pm 2.4$ | $66.2 \pm 1.4$ | $78.2 \pm 0.7$ | $88.8 \pm 0.5$ |
| AdaSubS | $0.0 \pm 0.0$ | $0.2 \pm 0.1$ | $8.6 \pm 0.4$ | $42.9 \pm 0.8$ | $65.6 \pm 1.0$ | $78.0 \pm 1.0$ |
| kSubS | $1.5 \pm 0.8$ | $19.1 \pm 5.7$ | $48.9 \pm 1.9$ | $55.9 \pm 1.3$ | $57.8 \pm 2.3$ | $64.3 \pm 3.3$ |
| HIPS-$\varepsilon$ | $\mathbf{23.6} \pm 2.2$ | $\mathbf{53.4} \pm 2.8$ | $\mathbf{76.2} \pm 2.3$ | $\mathbf{83.9} \pm 1.6$ | $\mathbf{90.6} \pm 1.0$ | $\mathbf{95.5} \pm 0.7$ |
| HIPS-$\varepsilon$-env | $5.7 \pm 0.8$ | $32.7 \pm 2.8$ | $65.8 \pm 1.8$ | $78.1 \pm 1.1$ | $85.9 \pm 0.9$ | $92.6 \pm 0.5$ |

| | Sliding Tile Puzzle | | | | | |
|---|---|---|---|---|---|---|
| $s$ | 2 | 5 | 10 | 20 | 40 | 60 |
| PHS* | $0.0 \pm 0.0$ | $\mathbf{98.8} \pm 0.2$ | $\mathbf{100.0} \pm 0.0$ | $\mathbf{100.0} \pm 0.0$ | $\mathbf{100.0} \pm 0.0$ | $\mathbf{100.0} \pm 0.0$ |
| HIPS | $0.0 \pm 0.0$ | $0.0 \pm 0.0$ | $0.2 \pm 0.1$ | $22.7 \pm 2.9$ | $70.9 \pm 4.0$ | $85.6 \pm 2.2$ |
| HIPS-env | $0.0 \pm 0.0$ | $0.0 \pm 0.0$ | $0.0 \pm 0.0$ | $14.4 \pm 2.0$ | $64.5 \pm 4.4$ | $83.2 \pm 2.6$ |
| AdaSubS | $0.0 \pm 0.0$ | $0.0 \pm 0.0$ | $0.0 \pm 0.0$ | $0.0 \pm 0.0$ | $0.0 \pm 0.0$ | $0.0 \pm 0.0$ |
| kSubS | $0.0 \pm 0.0$ | $0.0 \pm 0.0$ | $0.0 \pm 0.0$ | $58.2 \pm 5.3$ | $88.2 \pm 1.4$ | $89.9 \pm 1.4$ |
| HIPS-$\varepsilon$ | $0.0 \pm 0.0$ | $0.1 \pm 0.0$ | $18.1 \pm 3.1$ | $71.8 \pm 4.3$ | $95.2 \pm 1.3$ | $98.9 \pm 0.4$ |
| HIPS-$\varepsilon$-env | $0.0 \pm 0.0$ | $0.0 \pm 0.0$ | $9.9 \pm 1.2$ | $65.4 \pm 4.6$ | $93.9 \pm 1.6$ | $98.3 \pm 0.7$ |

| | Box-World | | | | | |
|---|---|---|---|---|---|---|
| $s$ | 1 | 2 | 5 | 10 | 20 | 60 |
| PHS* | $4.5 \pm 0.7$ | $9.3 \pm 1.1$ | $34.9 \pm 3.3$ | $\mathbf{73.6} \pm 2.9$ | $91.4 \pm 0.6$ | $92.4 \pm 0.5$ |
| HIPS | $0.0 \pm 0.0$ | $25.4 \pm 1.5$ | $38.0 \pm 3.1$ | $61.3 \pm 3.8$ | $88.9 \pm 2.1$ | $99.6 \pm 0.1$ |
| HIPS-env | $0.0 \pm 0.0$ | $\mathbf{28.6} \pm 1.8$ | $\mathbf{39.5} \pm 2.6$ | $63.4 \pm 3.3$ | $89.3 \pm 2.3$ | $99.7 \pm 0.1$ |
| HIPS-$\varepsilon$ | $\mathbf{5.9} \pm 0.7$ | $14.6 \pm 2.2$ | $30.0 \pm 3.5$ | $64.1 \pm 3.5$ | $93.4 \pm 1.1$ | $99.7 \pm 0.1$ |
| HIPS-$\varepsilon$-env | $5.3 \pm 0.5$ | $12.3 \pm 1.5$ | $28.1 \pm 3.2$ | $61.4 \pm 3.9$ | $\mathbf{94.2} \pm 1.2$ | $\mathbf{99.9} \pm 0.0$ |

| | Travelling Salesman Problem | | | | | |
|---|---|---|---|---|---|---|
| $s$ | 3.5 | 5 | 7.5 | 10 | 20 | 60 |
| PHS* | $0.3 \pm 0.1$ | $0.8 \pm 0.2$ | $10.6 \pm 3.4$ | $39.2 \pm 8.8$ | $87.7 \pm 3.5$ | $93.5 \pm 2.3$ |
| HIPS | $0.3 \pm 0.3$ | $19.0 \pm 6.0$ | $75.9 \pm 8.0$ | $97.6 \pm 1.0$ | $99.0 \pm 0.0$ | $\mathbf{100.0} \pm 0.0$ |
| HIPS-env | $8.8 \pm 3.6$ | $61.6 \pm 12.6$ | $98.5 \pm 0.4$ | $\mathbf{98.9} \pm 0.1$ | $99.5 \pm 0.1$ | $\mathbf{100.0} \pm 0.0$ |
| AdaSubS | $0.0 \pm 0.0$ | $0.0 \pm 0.0$ | $0.0 \pm 0.0$ | $0.0 \pm 0.0$ | $0.0 \pm 0.0$ | $0.0 \pm 0.0$ |
| kSubS | $0.0 \pm 0.0$ | $0.0 \pm 0.0$ | $0.0 \pm 0.0$ | $0.0 \pm 0.0$ | $0.0 \pm 0.0$ | $33.4 \pm 12.9$ |
| HIPS-$\varepsilon$ | $4.1 \pm 1.5$ | $49.2 \pm 9.0$ | $96.6 \pm 1.3$ | $98.8 \pm 0.1$ | $\mathbf{99.9} \pm 0.0$ | $\mathbf{100.0} \pm 0.0$ |
| HIPS-$\varepsilon$-env | $\mathbf{29.1} \pm 8.5$ | $\mathbf{90.0} \pm 3.9$ | $\mathbf{98.8} \pm 0.2$ | $\mathbf{98.9} \pm 0.0$ | $99.8 \pm 0.1$ | $\mathbf{100.0} \pm 0.0$ |

# K Environment Interactions

In addition to the number of search node expansions and running time, the search cost can be evaluated as a function of the low-level environment steps during the search. We compare HIPS-$\varepsilon$ with a learned environment model and guaranteed completeness (see Appendix E) to PHS*, kSubS, and AdaSubS and analyzed the percentage of puzzles solved given $N$ environment steps. We omitted HIPS from the comparison, as the variant of HIPS with environment dynamics is extremely wasteful with the environment steps, whereas HIPS with the learned models does not rely on the environment simulator at all, and we compared HIPS-$\varepsilon$ to HIPS in that setting in Table 2. We also included the results for HIPS-$\varepsilon$ when the number of simulator calls has been included in the search cost as the last row of the table. Our results show that HIPS-$\varepsilon$ is very efficient in terms of environment interactions, outperforming PHS* in every environment, even if the number of model calls is included in the search cost. kSubS is also highly wasteful with the low-level environment steps, whereas AdaSubS is highly competitive with HIPS-$\varepsilon$ in Sokoban but fails to perform well in STP or TSP due to the lower overall solution percentage.

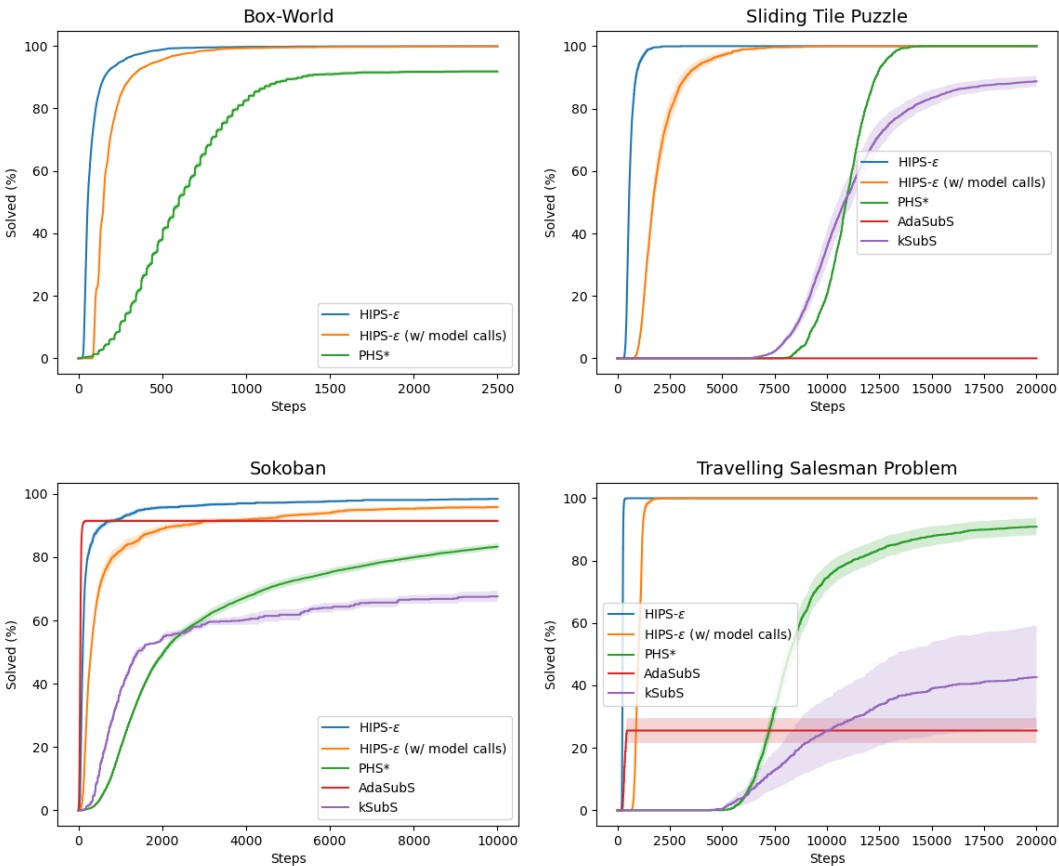

Figure 10: The mean percentage of puzzles solved (%) as a function of the number of environment steps for different search methods. For HIPS-$\varepsilon$, we have also plotted the solution percentage, assuming that each dynamics function call is equal to an environment step. The shaded area is one standard error. For HIPS-$\varepsilon$, we use the same values of $\varepsilon$ as in Table 9. We use PHS* as the underlying search algorithm in Sokoban, STP and Box-World, and A* in TSP.

Table 15: The mean success rates (%) after $N$ low-level environment steps during the search and the standard error of the mean as the uncertainty metric.

| | Sokoban | | | | | |
|---|---|---|---|---|---|---|
| $N$ | 50 | 100 | 250 | 1000 | 2500 | 10000 |
| PHS* | $0.0 \pm 0.0$ | $0.1 \pm 0.0$ | $0.6 \pm 0.1$ | $19.5 \pm 0.5$ | $56.5 \pm 1.3$ | $83.3 \pm 1.1$ |
| AdaSubS | $\mathbf{30.9} \pm 1.3$ | $\mathbf{60.3} \pm 0.7$ | $74.4 \pm 0.8$ | $85.2 \pm 0.9$ | $88.0 \pm 0.9$ | $90.2 \pm 0.9$ |
| kSubS | $0.0 \pm 0.0$ | $0.0 \pm 0.0$ | $1.7 \pm 0.5$ | $37.9 \pm 0.4$ | $56.0 \pm 1.1$ | $67.6 \pm 1.7$ |
| HIPS-$\varepsilon$ | $9.7 \pm 0.7$ | $47.4 \pm 2.3$ | $\mathbf{81.3} \pm 2.0$ | $\mathbf{92.2} \pm 0.8$ | $\mathbf{96.1} \pm 0.5$ | $\mathbf{98.4} \pm 0.2$ |
| HIPS-$\varepsilon$ (w/ model calls) | $0.0 \pm 0.0$ | $3.9 \pm 0.3$ | $42.4 \pm 2.7$ | $82.0 \pm 1.8$ | $90.1 \pm 1.0$ | $95.8 \pm 0.6$ |

| | Sliding Tile Puzzle | | | | | |
|---|---|---|---|---|---|---|
| $N$ | 500 | 1000 | 2500 | 5000 | 10000 | 20000 |
| PHS* | $0.0 \pm 0.0$ | $0.0 \pm 0.0$ | $0.0 \pm 0.0$ | $0.0 \pm 0.0$ | $20.6 \pm 0.5$ | $\mathbf{100.0} \pm 0.0$ |
| AdaSubS | $0.0 \pm 0.0$ | $0.0 \pm 0.0$ | $0.0 \pm 0.0$ | $0.0 \pm 0.0$ | $0.0 \pm 0.0$ | $0.0 \pm 0.0$ |
| kSubS | $0.0 \pm 0.0$ | $0.0 \pm 0.0$ | $0.0 \pm 0.0$ | $0.0 \pm 0.0$ | $35.6 \pm 5.4$ | $88.7 \pm 1.7$ |
| HIPS-$\varepsilon$ | $\mathbf{32.7} \pm 3.5$ | $\mathbf{93.1} \pm 1.7$ | $\mathbf{99.9} \pm 0.1$ | $\mathbf{100.0} \pm 0.0$ | $\mathbf{100.0} \pm 0.0$ | $\mathbf{100.0} \pm 0.0$ |
| HIPS-$\varepsilon$ (w/ model calls) | $0.0 \pm 0.0$ | $3.6 \pm 0.8$ | $78.9 \pm 4.1$ | $97.0 \pm 1.0$ | $99.9 \pm 0.1$ | $\mathbf{100.0} \pm 0.0$ |

| | Box-World | | | | | |
|---|---|---|---|---|---|---|
| $N$ | 50 | 100 | 250 | 500 | 1000 | 2500 |
| PHS* | $0.4 \pm 0.2$ | $1.4 \pm 0.2$ | $10.9 \pm 0.5$ | $38.1 \pm 1.0$ | $82.6 \pm 0.7$ | $91.8 \pm 0.5$ |
| HIPS-$\varepsilon$ | $\mathbf{40.1} \pm 1.0$ | $\mathbf{78.7} \pm 1.1$ | $\mathbf{94.9} \pm 0.3$ | $\mathbf{98.8} \pm 0.1$ | $\mathbf{99.7} \pm 0.0$ | $\mathbf{99.9} \pm 0.0$ |
| HIPS-$\varepsilon$ (w/ model calls) | $0.0 \pm 0.0$ | $16.4 \pm 0.4$ | $84.0 \pm 0.9$ | $95.5 \pm 0.3$ | $99.3 \pm 0.1$ | $99.8 \pm 0.0$ |

| | Travelling Salesman Problem | | | | | |
|---|---|---|---|---|---|---|
| $N$ | 250 | 500 | 1000 | 2500 | 10000 | 20000 |
| PHS* | $0.0 \pm 0.0$ | $0.0 \pm 0.0$ | $0.0 \pm 0.0$ | $0.0 \pm 0.0$ | $74.5 \pm 4.1$ | $90.9 \pm 2.7$ |
| AdaSubS | $0.0 \pm 0.0$ | $0.0 \pm 0.0$ | $3.3 \pm 1.6$ | $14.4 \pm 2.4$ | $24.4 \pm 3.3$ | $24.4 \pm 3.3$ |
| kSubS | $0.0 \pm 0.0$ | $0.0 \pm 0.0$ | $0.0 \pm 0.0$ | $0.0 \pm 0.0$ | $25.5 \pm 10.5$ | $42.6 \pm 16.4$ |
| HIPS-$\varepsilon$ | $\mathbf{44.9} \pm 4.4$ | $\mathbf{99.9} \pm 0.0$ | $\mathbf{100.0} \pm 0.0$ | $\mathbf{100.0} \pm 0.0$ | $\mathbf{100.0} \pm 0.0$ | $\mathbf{100.0} \pm 0.0$ |
| HIPS-$\varepsilon$ (w/ model calls) | $0.0 \pm 0.0$ | $0.0 \pm 0.0$ | $50.2 \pm 12.7$ | $99.9 \pm 0.0$ | $\mathbf{100.0} \pm 0.0$ | $\mathbf{100.0} \pm 0.0$ |