# OpenReview forum: "Hybrid Search for Efficient Planning with Completeness Guarantees"
_NeurIPS.cc/2023/Conference — NeurIPS 2023 poster_

### Official Review · Reviewer_i9rx · 2023-07-03

**Soundness:** 3 good
**Presentation:** 3 good
**Contribution:** 3 good
**Rating:** 7
**Confidence:** 3

**Summary:**

In this paper, the authors propose a hybrid technique to speed up the planning tasks. The novelty is that the completeness is guaranteed. I agree that guaranteeing completeness is a good property for a learning-based algorithm.

**Strengths:**

1. The paper is well-written.
2. The novelty of the contribution is moderate.
3. It seems that the algorithm tried to find a balance between classic algorithms (complete but slow) and learning-based algorithms (fast but non-complete). The simulated results look good.

**Weaknesses:**

1. I would say that the words "hybrid search" is not a suitable expression for the proposed algorithm. There have been too many "hybrid" algorithms. Currently I haven't come up with a suggestion, but the authors can think about it.
2. Can the author present the problem to be solved in a formal environment? I mean "Problem 1: (xxx) ...". Currently the presentation is only friendly to expert. (Fig.1 does a good job.)
3. I suggest the authors implement classic algorithms and show their performance. In my opinion, they are complete and optimal, but slow. By doing this the author can show the quality of the solution generated by the proposed algorithm.
4. The algorithms should be tested in larger simulated environments.
5. I think the related work section can be re-structured. Currently it just listed three kinds of searching algorithms, without mentioning their connection to this paper.

**Questions:**

Please see my comments in the "weakness" block.

**Limitations:**

I think the paper has presented the limitation and future works properly.

---

> ### Author Rebuttal · Authors · 2023-08-09
>
> We thank the reviewer for their insightful comments and questions.
>
> >I would say that the words "hybrid search" is not a suitable expression for the proposed algorithm. There have been too many "hybrid" algorithms. Currently I haven't come up with a suggestion, but the authors can think about it.
>
> Thank you for the comment. A more concrete alternative to hybrid search is a high-level search augmented with low-level actions. To make it easy for the potential reader, we introduce our main idea and present the name hybrid search already in the abstract.
>
> >Can the author present the problem to be solved in a formal environment? I mean "Problem 1: (xxx) ...". Currently the presentation is only friendly to expert. (Fig.1 does a good job.)
>
> We will add a more formal presentation of the problem, preferably at the beginning of Section 3, depending on the page limit constraints.
>
> >I suggest the authors implement classic algorithms and show their performance. In my opinion, they are complete and optimal, but slow. By doing this the author can show the quality of the solution generated by the proposed algorithm.
>
> We performed this analysis. Please see Table 12 in the common response pdf. Generally, we indeed note that HIPS-$\varepsilon$ greatly outperforms the classical planning algorithms in terms of node expansions, even if we allow classical planning algorithms to use heuristics defined using prior knowledge, which HIPS-$\varepsilon$ does not have access to.
>
> >The algorithms should be tested in larger simulated environments.
>
> We think that the environments used are already challenging and require significant computing power, as evidenced by the relative failure of classical planning methods at solving the problems (see Table 12 in the common response pdf). To further increase the difficulty, we perform the OOD experiments with Box-World (in the paper) and Sokoban (Figure 6 in the common response pdf). Also, the 5x5 sliding tile puzzle environment is considered challenging in the classical planning literature [1] (page 71).
>
> >I think the related work section can be re-structured. Currently it just listed three kinds of searching algorithms, without mentioning their connection to this paper.
>
> Thank you for this comment. The first class of algorithms is hierarchical planning methods, but they rely on numerical optimization or are not suited to solving difficult discrete problems as used for evaluation in this work due to lack of search, capability to generate exact subgoals, or ability to plan. The second class of algorithms represents an orthogonal direction for improving hierarchical planning algorithms, and successfully combining these algorithms with our work could lead to even stronger agents, which is a subject for further work. The third class of algorithms is the ones most closely related to our work that our work builds on and that we use as baselines for evaluating our method. We will incorporate this information in the related work section to clarify the connection to prior work.
>
> [1] Russell, S. & Norvig, P. Artificial Intelligence: A Modern Approach. 3rd edition.

---

> > ### Comment · Reviewer_i9rx · 2023-08-14
> >
> > First I would like to thank the authors for their comments. Please see my response to the authors' rebuttal.
> >
> > 1. "hybrid-search" issue: Sorry, I don't think "high-level search augmented with low-level actions" is an appropriate naming of the algorithm. And it is neither elegant to explain the paper title in abstract (then the title is useless). Can the author think more?
> >
> > 2. "experiment" issue: I still don't think the experimental settings are challenging given my knowledge in classic planning (now it is 2023!). I noticed that this issue was also raised by other reviewers.
> >
> > 3. "related work" issue: The authors tried to categorise and differ the existing algorithms, which is good. However, I hope the authors can explain more on this. For example, in the authors' response it is stated that the first class of algorithms cannot generate exact subgoal (what does it mean?), or ability to plan (but A* was presented in this category in the manuscript, isn't it?). In fact, after re-reading the manuscript, I think the arrangement of the related work section is not correct: It describes three classes of algorithms with different functionalities, instead of describing three classes of algorithm that tried to solve THE problem, "efficient planning on grids".

---

> > > ### Author Response · Authors · 2023-08-15
> > >
> > > Thank you for your additional comments and questions
> > >
> > > >"hybrid-search" issue: Sorry, I don't think "high-level search augmented with low-level actions" is an appropriate naming of the algorithm. And it is neither elegant to explain the paper title in abstract (then the title is useless). Can the author think more?
> > >
> > > We believe that the name Complete Subgoal Search (CSS/CSubS) would suit the approach. It highlights that the primary driver of the search is the subgoals, but the search is also made complete by including the low-level actions. Complete Subgoal Search with Low-level Actions, or CSSLA is also an alternative that includes the low-level actions in the name of the proposed search approach. We thank the reviewer for the suggestion and are happy to ponder this further. We are surely interested in using the best possible title.
> > >
> > > > "experiment" issue: I still don't think the experimental settings are challenging given my knowledge in classic planning (now it is 2023!). I noticed that this issue was also raised by other reviewers.
> > >
> > > We selected the experimental setup to validate the claims made in our paper. We have evaluated Complete Subgoal Search on all problems used in the original HIPS paper and showed that our method either outperforms or is on par with the baseline HIPS on each benchmark. Additionally, we have performed difficult OOD experiments to evaluate the transferability of our method. The results show significant improvements in comparison with prior work. Furthermore, we have compared the performance of our method to classical planning approaches, analyzed the sensitivity of our framework to the value of $\varepsilon$, and performed additional experiments to understand how the value of $\varepsilon$ affects the search behavior. We are happy to consider running additional experiments proposed by the reviewer if any of the claims in our paper is not validated by our experimental results.
> > >
> > > > "related work" issue: The authors tried to categorise and differ the existing algorithms, which is good. However, I hope the authors can explain more on this. For example, in the authors' response it is stated that the first class of algorithms cannot generate exact subgoal (what does it mean?), or ability to plan (but A* was presented in this category in the manuscript, isn't it?). In fact, after re-reading the manuscript, I think the arrangement of the related work section is not correct: It describes three classes of algorithms with different functionalities, instead of describing three classes of algorithms that tried to solve THE problem, "efficient planning on grids".
> > >
> > > By referring to an inability to generate exact subgoals, the models either do not generate subgoals at all or generate subgoals in the latent state, which may not correspond to valid or reachable states, which is an issue in environments such as Sokoban that require precisely correct actions to solve the tasks. Referring to a lack of ability to plan was an oversight on our part. Thank you for pointing that out. We will replace that with  “do not plan with an explicit search,” as CEM-based methods plan but do not perform a systematic search. We note that methods without search struggle to solve the benchmarks used in our work (see the original HIPS paper and Table 2). Furthermore, A* is indeed an algorithm that does have the ability to plan, but it is not a hierarchical algorithm. Note that we discussed A* and Dijkstra in the context of hierarchical planning algorithms that use them as subcomponents for search in a continuous or visual setting, which is different from ours.
> > >
> > > All in all, in the related work section, we have mostly presented and compared our approach with other hierarchical planning approaches suited for solving various families of problems, as we consider those algorithms to be the most closely related to ours from a methodological standpoint, even though the problems they solve are somewhat different with the exception of kSubS, AdaSubS, and HIPS. We are also happy to take suggestions if the reviewer believes we have missed some relevant line of work that should be discussed in our paper. Furthermore, to make the related work section easier to read we will add a short introduction to the beginning of the related work section that explains how we selected the different method categories and how the discussed methods relate to our approach.

---

> > > > ### Comment · Reviewer_i9rx · 2023-08-17
> > > >
> > > > Thank you for the feedback.
> > > >
> > > > 1. Naming issue: OK, at least they are much better than "hybrid search". I'm OK with either. As my final remark, the authors should be aware that when seeing "complete subgoal research" (type of things) the readers do not know the merit of the proposed algorithm. In other words, such descriptions only avoid confusion (which should not have been existed), do not really contribute to the clarity of the paper.
> > > >
> > > > 2. Related works & Experimental settings: I realise that they are because of the gap between different areas. The authors were trying to span the contribution across both learning-based planning and classic planning, but failed to provide a succinct literature review and a persuasive experimentation. For instance, A* is not hierarchical, but this is not a disadvantage of A*, so how did A* inspire the proposed algorithm? As the author mentioned, the experiments have been better than others in some benchmarks, then the authors can safely RESTRICT THE SCOPE OF THE PAPER WITHIN THIS AREA, i.e., removing possible connections/comparisons to classic algorithms. This will make the manuscript a solid one in its own research area.

---

> > > > > ### Author Response · Authors · 2023-08-18
> > > > >
> > > > > Thank you for your reply. Given our discussion, we are happy to follow your advice on carefully positioning the paper as a hierarchical planning method. We want to stress that we believe that our contribution is in the field of learning-based planning methods, and we want to ascertain that no misunderstanding about the scope of our work can occur by ensuring that the related work and experimental section fully reflect the positioning of our paper as a hierarchical, learning-based planning method.
> > > > >
> > > > > Specifically, we want to add the following at the beginning of the “Related Work” section: “This paper proposes to augment the high-level search used in learning-based hierarchical planning with low-level actions to achieve completeness while even improving the search performance in terms of node expansions. First, we discuss our work in the general context of hierarchical planning and then focus on the closest methods in discrete subgoal search, against which we also empirically evaluate the proposed approach.”
> > > > >
> > > > > We will also improve the main text of the “Related Work” section by clarifying how each mentioned hierarchical method is related to ours. Finally, we will remove the ambiguous mention of A* and Dijkstra so that no misunderstandings about the relationship of our work to classic planning can occur.
> > > > >
> > > > > In the experimental section, we will focus solely on the included comparisons to prior learning-based planning methods (Tables 1-3) and analysis and ablations of the proposed algorithm (Figures 3-4, Table 4). We added Table 12 to the response pdf to address the questions of the reviewers i9rx (your question in the original review of the classic planning performance) and nqzu, as the table can provide additional information about the benchmarks to readers with different backgrounds and may anchor the research to their previous knowledge. However, we agree that we need to be careful when positioning the paper so that no misunderstanding about the scope of our contribution (learning-based planning, not classic planning) can occur. Therefore, we are willing not to add Table 12 to the paper. We are also happy to consider further modifications to highlight the strengths of the paper.
> > > > >
> > > > > We are grateful that you see our paper as a solid contribution to the area of hierarchical planning methods.

---

### Official Review · Reviewer_jgQs · 2023-07-04

**Soundness:** 3 good
**Presentation:** 3 good
**Contribution:** 3 good
**Rating:** 6
**Confidence:** 3

**Summary:**

The paper introduces a method, called hybrid search (or HIPS-\epsilon) that combines high-level planning (with subgoals generated by a learned model) with low-level search. Subgoals allow for more efficient search, but existing subgoal-based methods are prone to errors, which can lead to failures in finding solutions, even if the solution exists. Low-level search gives the completeness guarantee but is usually much less efficient. Hybrid search is a novel and efficient way of combining the advantages of both approaches. The method was tested on 4 different tasks, and various analyses show its excellent performance.

**Strengths:**

The motivation for this paper is very clear and it precisely formulates the problem it is going to solve. The introduction section is well written with clear references to other works in the field. Baselines are chosen accurately. There are also a lot of insightful comments about the comparison of hybrid search to baselines. There is a broad evaluation of the method: success rate across all envs, epsilon-hyperparameter analysis, unsolved puzzles ratio, etc. Figure 3 is very insightful. I agree with the authors that OOD generalization is a very attractive property of the method. Hybrid search is explained in a clear way and, most importantly, the meaning of epsilon is easy to understand. The technical improvement over HIPS is simple, it mainly consists of two contributions: modification of policy, and modification of search heuristics. I consider it a huge advantage that a simple technique gives excellent results. The quality of the text is high, also I really liked all the comments. The main result, that is, achieving completeness without loss of efficiency, is excellent.

**Weaknesses:**

Table 1: there is no information on how many problem instances the method was tested. Are all results statistically significant? I know that there is no place in this table to put all error bars, but in the table caption, at least the average error (or maximum error) should be mentioned. I have looked into the supplementary materials to check the error estimates and I am not convinced by the results for TSP. For n=20, 50, and 100 the difference between HIPS and HIPS-epsilon is much smaller than the error estimates. Could you run the test on more instances to reduce the error or provide some argument why the results for TSP are meaningful?

Figure 1. does not serve its role in illustrating the method. A reader wants to look at the image and quickly see the difference between the hybrid search and the other methods. The whole figure does not help in it. It illustrates only some flaws of other methods. I was unable to get any idea what a hybrid search is about just by looking at Figure 1. I think that a much better Figure can be created instead.

Figure 3. Lines for epsilon=1e-5 are missing. You use that value in two problems, so we would like to see how it behaves on this graph.

There is no information about the size of the dataset used for offline training.

OOD Generalization should be more elaborated. It is a very interesting result. I would like to see more results, for example, a table similar to  Table 1, Table 2, or Figure 3. Please consider this a minor weakness, I don't see any flaw in the part of the paper about OOD, I just want to say that the paper could benefit much from getting more results like this.

The success rate is clearly greate


**Questions:**

Questions:
1. Why the error estimates for the success rate on the TSP problem are so high compared to the other tasks?
2. How sample-efficient is the training of hybrid search? Is there any way to estimate the number of samples needed to train the method in a new env?
3. How the performance of hybrid search depends on the size of the training dataset?
4. Clearly TSP is an outlier: the results on this task look different (I mean: Figure 3, Table 7 in supplementary materials or lines 261-262). Do you know why it is so? What is so specific about TSP?
5. Do you have results similar to Table 3 but for other problems? The most interesting here is the average number of nodes of expansion.
6. A typical solution produced by HIPS-epsilon was constructed from some number of high-level actions and some number of low-level actions. What is the ratio of those? How is depends on epsilon?
7. What happens if you run HIPS-epsilon on a problem with no solution (e.g. unsolvable Sokoban board)? For example, the state space in Sokoban is finite. Could HIPS be used to classify if the solution exists? Sokoban is an interesting example since deciding if a given board is solvable is NP-hard problem.
8. Was AdaSubS tuned for the experiments? Its performance strongly depends on the chosen hyperparameters.

lines: 269-272: was any component of HIPS or HIPS-epsilon trained after modification of the dataset? I suppose that no, just want to be sure.

Suggestions:
I think you should mention the tasks on which the method is evaluated both in the abstract and introduction. It is important for the reader.

In Section 3.1 some comment about the meaning of the heuristic factor is missing. For a reader who is not familiar with PHS it may be hard to quickly get how it depends on h(n) and we need the heuristic factor at all. I know that this can be found in the cited papers. I only suggest adding a footnote or comment with some motivation or explanation.


**Limitations:**

Some limitations were mentioned in Section 5, but in my opinion not all. All problems used for testing HIPS-epsilon have compact state representation, finite action space (there are problems with discrete yet infinite action space). Also, HIPS-epsilon is useful only on problems for which the solution exists (it is not a problem, but should be mentioned).

---

> ### Author Rebuttal · Authors · 2023-08-09
>
> We thank the reviewer for their insightful comments and questions.
>
> >Are all results statistically significant?
> >Could you run the test on more instances to reduce the error or provide some argument why the results for TSP are meaningful?
>
> We will incorporate the information about statistical significance in the main text. At the moment, the statistical significance of the results has been computed in the supplementary material. The results are statistically significant in almost all cases except for TSP. Note that for TSP, we expect HIPS-$\varepsilon$ and HIPS to be equal given that HIPS already attains a 100 % success rate and low-level expansions are never performed. The difference is not expected to be statistically significant.
>
> >Figure 1. does not serve its role in illustrating the method.
>
> We will do our best to improve the figure for the final version of the paper by clearly marking and annotating the figure, illustrating the search queue, and reducing the number of nodes, in case the paper is accepted.
>
> > Figure 3. Lines for epsilon=1e-5 are missing. You use that value in two problems, so we would like to see how it behaves on this graph.
>
> The results for epsilon = 10^-5 are very close to those of epsilon = 10^-3, so we chose to omit 10^-5 for clarity. We will include a figure with epsilon = 10^-5 in the final version of the paper if accepted.
>
> > There is no information about the size of the dataset used for offline training.
>
> We use the same dataset as in [1], that is, 10340 trajectories in Sokoban, 5100 in STP, 22100 in BW, and unlimited but extremely low-quality trajectories in TSP.
>
> >OOD Generalization should be more elaborated.
>
> We added a similar experiment on Sokoban (see Figure 6 in the common PDF), and add numerical results in the appendixes (due to lack of space).
>
> > Why the error estimates for the success rate on the TSP problem are so high compared to the other tasks?
>
> The performance of HIPS-$\varepsilon$ in TSP is more variable than in other environments. We hypothesize that it is due to the environment being particularly sensitive to the successfulness of the segmentation (does the detector agent always learn to take every city visit as a subgoal).
>
> >Sample efficiency
>
> In this work, we used datasets of fixed sizes, and HIPS-epsilon outperformed the baselines for all sizes, but we do not unfortunately have any general estimates. However, note that the sample efficiency of the overall method depends on the chosen subgoal search approach, and in particular, the generative model (in this case, HIPS, and VQVAE). Generally, our results indicate that the hybrid search proposed by us will improve the sample efficiency of the underlying subgoal search (that is, HIPS-$\varepsilon$ will be better in this respect than HIPS), as it will enable the agent to deal with the imprecisions of the low-level policy and generative model, similarly as in the OOD experiments. Table 3 in our submission indicates that the sample efficiency can improve even if the problem is solvable with subgoal search.
>
> >What is so specific about TSP?
>
> The main difference is that HIPS already solves 100 % of the environments, so there is no room for improvement. In general, as mentioned above, we think that the environment makes successful segmentation particularly important, and there is some variance in terms of the performance of the detector agent trained with REINFORCE.
>
> > Do you have results similar to Table 3 but for other problems? The most interesting here is the average number of nodes of expansion.
>
> We do not unfortunately have the same results for other problems, as we sampled the evaluation environments in the environments. However, in Box-World, the average number of expansions is 3.56 vs 3.82 in favor of HIPS-$\varepsilon$ (p<0.001) for all solved problems. In Sokoban and TSP, we would expect the averages to be approximately equal for problems solved by both methods (given that $\varepsilon \to 0$ seems to work the best).
>
> > A typical solution produced by HIPS-epsilon was constructed from some number of high-level actions and some number of low-level actions. What is the ratio of those? How is depends on epsilon?
>
> Please see Figure 7 in the common pdf. High-level actions dominate the solutions, with individual low-level actions being used (between 15 % and 0.2 % in our experiments), and their share depends on the value of epsilon such that higher epsilon gives more low-level actions, as expected.
>
> >What happens if you run HIPS-epsilon on a problem with no solution (e.g. unsolvable Sokoban board)? For example, the state space in Sokoban is finite. Could HIPS be used to classify if the solution exists? Sokoban is an interesting example since deciding if a given board is solvable is NP-hard problem.
>
> Unfortunately not by any other way except by running the search until the search queue empties. This is an interesting topic for further work.
>
> >Was AdaSubS tuned for the experiments?
>
> To save computational resources, we copied the results for AdaSubS from [1], who did not elaborate in their paper on how the AdaSubS hyperparameters were chosen. In this paper, we incorporated our search approach into HIPS yielding HIPS-$\varepsilon$. A further interesting evaluation could be incorporating hybrid search into AdaSubS yielding AdaSubS-$\varepsilon$. In this case, we could also check hyperparameter optimization for AdaSubS.
>
> >lines: 269-272: was any component of HIPS or HIPS-epsilon trained after modification of the dataset? I suppose that no, just want to be sure.
>
> No, no retraining was performed.
>
> >Suggestions:
> >Some limitations were mentioned in Section 5, but in my opinion not all.
>
> Thank you for the suggestions and pointing out limitations, we will try to make them fit in the main text while staying within the constraints of the page limits.
>
> [1] Kujanpää, K., Pajarinen, J., & Ilin, A. (2023). Hierarchical Imitation Learning with Vector Quantized Models. ICML 2023.

---

> > ### Comment · Reviewer_jgQs · 2023-08-16
> > **Thanks**
> >
> > Thank you for all the answers.

---

### Official Review · Reviewer_v7TG · 2023-07-04

**Soundness:** 3 good
**Presentation:** 3 good
**Contribution:** 2 fair
**Rating:** 7
**Confidence:** 3

**Summary:**

The authors present an idea of enriching a classical hierarchical search pipeline with an exhaustive low-level search. This approach guarantees the completeness of the search and offers practical advantages, including slightly better success rates in the tested environments and stronger out-of-distribution evaluation properties. The method is built on top of the HIPS algorithm. The paper provides technical adjustments to the theory of the original HIPS that cover the hybrid approach.

**Strengths:**

The paper is mostly clear and well-written. The main idea is intuitive and the reported experiments support it. The OOD application seems very promising to me.

**Weaknesses:**

The novelty of the approach is limited, although the paper may still be a good contribution. I am not convinced that the completeness is a major concern itself, the paper lacks a clear justification for that. The impact of tuning the most important parameter $\varepsilon$ on the number of low-level expansions should be discussed. The key parameters used for evaluating presented methods should be provided.

**Questions:**

In general, I think that the paper is solid and well-presented. The novelty is limited, as the main idea is a formalized study of ideas hinted in some previous papers, which is an extension of known algorithms. Nevertheless, I think it may be a good contribution since the described approach is simple, yet offers some clear advantages. That said, I would like a few clarifications. See my comments below.

l.15: You claim that your approach guarantees completeness, which is clear. However, it is not clear to me why should we care so much about completeness. I suggest adding to the paper (introduction perhaps) a short justification of the necessity of having the completeness property, as you claim it to be your main advantage.

l.92 How much data do you use for training in a single environment? How long do you train each of the networks?

l.141: I understand that the value of $\varepsilon$ corresponds to the density of using low-level actions. But I would like to understand how the specific values (importantly, those used in experiments) correspond to the number of low-level expansions performed during the search. For instance, does setting $\varepsilon=10^{-5}$ correspond to roughly one expansion every $10^5$ steps? I don't think so, because it would have quite a negligible impact on performance. In l.212 you claim that a low value of $\varepsilon$ is generally preferred, but I don't know how it exactly relates to the search itself.

l.173: Please refer to the exact place in the appendix.

Table 1: Is the $\infty$ budget a theoretical bound (like we're sure that given $10^{20}$ iterations low-level search would solve TSP optimally), or did you simply run the methods for a very large number of steps? Please state it clearly.

Table 1: Since setting $\varepsilon\to 0$ essentially means that you perform exhaustive expansion in case the search would otherwise fail, why does your HIPS-e achieve (a little) worse results than HIPS in TSP?

Table 1: Please provide the values of the main hyperparameters used for evaluating all the approaches presented in Table 1. Did you tune any values yourself? The results of the AdaSubS baseline on Sliding Tile Puzzle and TSP seem quite low compared to kSubS. As far as I understand, it is a generalization of kSubS, so why is it so much worse?

l.259: Arguably the simplest approach to improving the completeness is to increase the number of subgoals generated at each node expansion. I wonder how do your empirical results relate to tuning that parameter? In particular, please specify the values that you use.

l.266 I really like the OOD idea. Intuitively, it seems clear that augmenting the search with reliable low-level expansions is helpful in case the generator struggle in unknown domains. I think it deserves analyzing it in more detail. Did you observe similar patterns in other environments? What budget did you use for the reported results? Or are these results _theoretical_ (i.e. HIPS-e is _guaranteed_ to solve everything eventually) (in which case you should also remark it)?

**Limitations:**

The limitations are discussed. The negative societal impact is not a concern here.

---

> ### Author Rebuttal · Authors · 2023-08-09
>
> We thank the reviewer for their insightful comments and questions.
>
> >I am not convinced that the completeness is a major concern itself.
> >
> > l.15 Why should we care so much about completeness
>
> We see completeness as a worthwhile problem to tackle due to four main reasons:
> 1. Without completeness, we do not know whether a solution will be found when the algorithm is executed. This is critical for both theoretical science but also for practical algorithms.
> 2. Completeness is the key to the promising OOD capabilities shown by our agent, as completeness guarantees that the solution to the tasks will be found, even if the generative model or low-level policy is imperfect. Completeness allows extensions and new incremental algorithms that require building on top of exact solutions. One example of that could be curriculum learning.
> 3. Achieving completeness makes applying high-level search as an alternative to low-level search possible in safety-critical real-world systems.
> 4. Without completeness, comparing different high-level search algorithms requires a somewhat arbitrary balancing of efficiency and solution percentage.
>
> We will include this discussion in the final version.
>
> > The impact of tuning epsilon
> >
> > l.141: How the specific values [of epsilon] correspond to the number of low-level expansions
>
> We performed an additional study on STP and analyzed the impact of $\varepsilon$ on the number of low-level expansions performed during the search and low-level actions included in the discovered solutions. Please see Figure 7 in the global response PDF for the results. As expected, the relative share of low-level expansions and actions decreases as we decrease epsilon, and the function is monotone. Note that $\varepsilon$ only affects the probability assigned to the node ($\pi(n)$ in Eq 4 of the paper), but the node evaluation function also depends on the node’s depth and low-level distance from the root, and the value of the learned heuristic function. Hence, there is no deterministic correspondence between the value of epsilon and the share of low-level expansions. Nevertheless, we observe that for $\varepsilon = 0.5$, the share of low-level expansions is slightly over 40 %, so there is a rough correspondence for larger values of $\varepsilon$. In line 212, we hypothesize that a low value of $\varepsilon$ should, in most cases, lead to a more efficient search due to high-level actions being used more often, but it has a worse worst case-performance.
>
> > Key parameters used for evaluating presented methods.
>
> We will add the hyperparameters used for evaluating HIPS and HIPS-epsilon to the final version. Note that for AdaSubS, kSubS, and other baselines, we copied the results from [1] (lines 238-239 in the submission) to save computational resources. We used the same hyperparameters for HIPS as in [1]. The only hyperparameter that we tuned for HIPS-$\varepsilon$ was the value of $\varepsilon$. AdaSubS has a learned low-level policy, whereas kSubS uses a low-level search. Therefore, AdaSubS solves a more difficult problem than kSubS (and equally difficult as HIPS-$\varepsilon$). The authors of [1] report that AdaSubS struggles to reliably reach the generated subgoals with the learned low-level policy, which would explain the results.
>
> > l.92 Training data and procedure
>
> We use the same dataset as in [1], that is, 10340 trajectories in Sokoban, 5100 in STP, 22100 in BW, and unlimited but extremely low-quality trajectories in TSP. The datasets used in [1] also contain a validation set. We used the validation loss for early stopping.
>
> > l.173: Exact place in the appendix.
>
> Appendix B, we will add it to the text.
>
> > Table 1. Is the budget a theoretical bound
>
> For HIPS-$\varepsilon$, a budget of 10,000 expansions was sufficient to solve all the problem instances in our experiments in Table 1. For Sokoban and STP, we ran a PHS* low-level search for all problems until all solutions were discovered. For BW and TSP, we had a computation budget of 128,000 expansions, and we needed to interrupt the PHS* evaluation runs before a 100 % solution rate was attained.
>
> >Table 1: Why does your HIPS-e achieve (a little) worse results than HIPS in TSP?
>
> The difference between HIPS and HIPS-$\varepsilon$ in TSP is just noise (please check the complete results in Appendix E, Table 7). Thank you for raising this question, we will clarify it in the main text.
>
> > l. 259. Number of subgoals
>
> Unfortunately, we did not have the computational resources for an in-depth analysis of how the number of subgoals generated at each node expansion affects the completeness. However, looking at Figures 6b, 7b, 8b, and 9b in the Appendix J of [1], it seems that the VQVAE is already "saturated" and increasing the number of VQVAE codes does not increase the number of generated subgoals. For instance, in Figure 7b, it appears as if the VQVAE generated 4, 4, and 7 valid subgoals for the given states, even though the requested number of subgoals was 64. Hence, given a VQVAE generative model, increasing the number of subgoals is very unlikely to lead to completeness. For the autoregressive network used in kSubS and AdaSubS, that could be verified separately in the future.
>
> > l.266 OOD idea
>
> We observed a similar pattern in Sokoban, although the advantage of HIPS-$\varepsilon$ is slightly smaller due to HIPS generalizing better than in Box-World. Please see Figure 6 in the common PDF for the results. In STP, generating OOD puzzles is impossible without increasing the board resolution, which would require re-training the networks, and in our preliminary experiments, we found that HIPS already generalizes well to increasing the number of cities in TSP, so the benefits of hybrid search are limited there. The budget for HIPS-$\varepsilon$ was 20,000 expansions, which was sufficient for solving all the evaluated problems.
>
> [1] Kujanpää, K., Pajarinen, J., & Ilin, A. (2023). Hierarchical Imitation Learning with Vector Quantized Models. ICML 2023.

---

> > ### Comment · Reviewer_v7TG · 2023-08-10
> >
> > Thank you for the clarifications. I acknowledge the importance of guaranteeing completeness, and I am pleased that you will include the justification.
> >
> > > The impact of tuning epsilon
> >
> > Thank you for the analysis, I think it's a nice insight into what's happening under the hood. Although it would be better to choose an environment where HIPS-$\varepsilon$ shows a greater advantage over HIPS than in TSP, in order to better understand the relation between low-level expansions and performance advantage. I don't demand providing another chart in this discussion, but I suggest adding it to the paper in later revisions. It looks like the benefit of using low-level expansions is related to the complexity of proposing a valid high-level subgoal (hence, more important in STP). Is that true?
> >
> > > Table 1: Why does your HIPS-e achieve (a little) worse results than HIPS in TSP?
> >
> > If it is just noise, I'm not sure if such an environment is relevant to this paper. I understand that you use it just because it was used in [1].
> >
> > > l. 259. Number of subgoals
> >
> > Please explain how you handle invalid subgoals. In particular, how do you count them into the search budget? If you request 60 subgoals out of which 7 turn out to be valid, does it count as roughly 7 or 60 when calculating the budget for Table 1? It seems to me that you consider it as a single expansion, correct me if I'm wrong. That's unfortunate since it is way way more costly than a single step of low-level search. To be fair, I think you should include in the search budget the low-level steps used to verify the subgoals, both valid and invalid. Also, you should include running time comparisons between the methods. At least a simple mean running time, at least in the appendix.

---

> > > ### Author Response · Authors · 2023-08-11
> > >
> > > Thank you for your additional comments and questions.
> > >
> > > >Although it would be better to choose an environment where HIPS-$\varepsilon$ shows a greater advantage over HIPS than in TSP, in order to better understand the relation between low-level expansions and performance advantage.
> > >
> > > Just to clarify, we chose to perform the analysis on STP (Sliding Tile Puzzle), not TSP (Traveling Salesman Problem). Sliding Tile Puzzle is the environment where HIPS-$\varepsilon$ shows the greatest performance advantage over HIPS. We will write the environment name out explicitly in the caption of Figure 6 to avoid this confusion in the future.
> > >
> > > >It looks like the benefit of using low-level expansions is related to the complexity of proposing a valid high-level subgoal (hence, more important in STP). Is that true?
> > >
> > > We agree.
> > >
> > > >If it is just noise, I'm not sure if such an environment is relevant to this paper. I understand that you use it just because it was used in [1].
> > >
> > > We agree, which is why we did not focus too much on it in our discussion. However, we included TSP for two reasons:
> > > 1. We did not want to cherry-pick environments from [1] and only select those where our method is useful
> > > 2. It illustrates that by choosing a suitable strategy ($\varepsilon \to 0$), our hybrid search has no disadvantages over HIPS even if the hybrid approach is not necessary, and even if a sub-optimal value of $\varepsilon$ is used (see Figure 3), the loss in performance is reasonable.
> > >
> > > >Please explain how you handle invalid subgoals. In particular, how do you count them into the search budget? If you request 60 subgoals out of which 7 turn out to be valid, does it count as roughly 7 or 60 when calculating the budget for Table 1? It seems to me that you consider it as a single expansion, correct me if I'm wrong. That's unfortunate since it is way way more costly than a single step of low-level search. To be fair, I think you should include in the search budget the low-level steps used to verify the subgoals, both valid and invalid. Also, you should include running time comparisons between the methods. At least a simple mean running time, at least in the appendix.
> > >
> > > Your understanding of how we count the search cost is correct: a node expansion always incurs a cost of one. The number of search node expansions has been used as the evaluation metric in prior work on subgoal search (kSubS, AdaSubS, HIPS), and we chose not to deviate from that. All high-level search methods that we used as baselines perform very many low-level environment steps per search node expansion, which is why we believe that the comparison to other high-level search methods is fair. For instance, kSubS uses a low-level search to verify the subgoals, which requires a substantial number of low-level environment steps.
> > >
> > > When comparing against low-level search methods, one node expansion is naturally more expensive. If the number of low-level environment steps needed for verifying the subgoals is used as the search cost, one high-level expansion can be roughly 500-1000 times as expensive as a low-level expansion in the worst-case scenario, depending on the environment and number of duplicate subgoals. This factor can be significantly reduced by parallelizing the subgoal verification. We believe that relying on a learned dynamics model to prune infeasible subgoals would also work for reducing the environment steps. However, HIPS-epsilon significantly outperforms low-level search in terms of node expansions, almost always with more than the factor of 1000 (see Table 12, for example). Thus, combining these two factors (and ignoring the benefits of parallelization), our method is superior to the low-level search in Sokoban, Box-World, and TSP. In the Sliding Tile Puzzle, where defining a suitable heuristic with prior knowledge is easy, our method is roughly on par with the low-level search in terms of the environment steps (depending on the value of W). We will modify Table 12 accordingly and include the running time comparisons in the appendix.

---

> > > > ### Comment · Reviewer_v7TG · 2023-08-14
> > > >
> > > > > Just to clarify, we chose to perform the analysis on STP (Sliding Tile Puzzle), not TSP (Traveling Salesman Problem).
> > > >
> > > > Right, sorry for the mistake. Now it makes much more sense. Having said that, from the charts I understand that the low-level expansions are roughly 0.2% (once every 500 expansions) for $\varepsilon=10^{-5}$, used in TSP. How can you solve so much more instances (twice more with 50 expansions) with such a tiny intervention (roughly one low-level expansion for 10 episodes, which on an unreasonable extreme can give +10% of success rate)? Can you somehow explain that or correct me?
> > > >
> > > >
> > > > > The number of search node expansions has been used as the evaluation metric in prior work on subgoal search (kSubS, AdaSubS, HIPS).
> > > >
> > > > That is actually not true. I see that AdaSubS defines the budget as a sum of high-level and low-level steps (see Appendix C in [1]). And I think it makes more sense since it is better correlated with the actual complexity (which is basically what we care about). Adding such comparison would make the paper stronger.
> > > >
> > > > > All high-level search methods that we used as baselines perform very many low-level environment steps per search node expansion
> > > >
> > > > Again, since AdaSubS use fixed-length subgoals, it also makes a limited number of low-level steps (if I understand correctly, it is the value of $C_2$, see Appendix F in [1]). It seems to be ~10, which is way more than 1, but also way less than 500-1000.
> > > >
> > > >
> > > > [1] Zawalski, Tyrolski, Czechowski, Odrzygóźdź, Stachura, Piękos, Wu, Kuciński, Miłoś; Fast and Precise: Adjusting Planning Horizon with Adaptive Subgoal Search

---

> > > > > ### Author Response · Authors · 2023-08-15
> > > > >
> > > > > Thank you for your additional comments and questions
> > > > >
> > > > > >How can you solve so much more instances (twice more with 50 expansions) with such a tiny intervention (roughly one low-level expansion for 10 episodes, which on an unreasonable extreme can give +10% of success rate)? Can you somehow explain that or correct me?
> > > > >
> > > > > Thank you for your question. We assume that you are referring to Table 3 in the paper. One low-level expansion for every 500 expansions would mean that for 10 trajectories at 50 expansions, one of the ten trajectories has benefitted from an additional low-level expansion, which could, in theory, exactly correspond to the 10 % difference that we observe in the results. However, this calculation assumes that the low-level expansions are evenly distributed between the start and end of the search. Actually, the greatest benefit of low-level expansions occurs at the beginning of the search. For $\varepsilon = 10^{-5}$ in STP, almost every (>98 %) low-level expansion is performed early in the search (before 10 expansions), including the problem instances which the high-level search can solve without relying on low-level actions. Therefore, we hypothesize that the hybrid search discovers some valuable subgoals that lead to a solution quickly and that a high-level-only search would have missed otherwise. We believe that this explains the surprisingly big difference in Table 3. Note also that the first few subgoals are the most critical ones. The most common failure case of HIPS in STP is that the search fails immediately, as HIPS does not generate sufficiently many valid subgoals (see Figure 7B in the original paper).
> > > > >
> > > > > >That is actually not true. I see that AdaSubS defines the budget as a sum of high-level and low-level steps (see Appendix C in [1]). And I think it makes more sense since it is better correlated with the actual complexity (which is basically what we care about). Adding such comparison would make the paper stronger.
> > > > >
> > > > > >
> > > > > >
> > > > > >Again, since AdaSubS use fixed-length subgoals, it also makes a limited number of low-level steps (if I understand correctly, it is the value of $C_2$, see Appendix F in [1]). It seems to be ~10, which is way more than 1, but also way less than 500-1000.
> > > > >
> > > > > Thank you for pointing that out. We had missed that and apologize for our previous, incorrect answer. Your argument is valid. AdaSubS performs approximately 10 low-level steps per generated subgoal, multiplied by the number of subgoals generated, which, looking at Table 12 in Appendix H, seems to be one for each of the three generative networks, which makes AdaSubS efficient in terms of low-level environment steps. When executed naïvely or assuming that the number of low-level environment steps does not matter, HIPS, and consequently, HIPS-$\varepsilon$, would perform 10-40 low-level environment steps per generated subgoal, which would lead to 500-1000 low-level steps per expansion. Note that the inefficiency in low-level environment steps is primarily a property of HIPS when the true environment dynamics are used, not something that our hybrid search approach, and the corresponding HIPS-$\varepsilon$, have led to.
> > > > >
> > > > > However, using a learned dynamics model allows us to significantly reduce the number of low-level environment steps. We have already evaluated this in Table 2 of the paper, and the results show that if the learned dynamics model is accurate, HIPS-$\varepsilon$ can solve puzzles without interacting with the environment at all, but the completeness guarantee is lost if the environment dynamics are inaccurate. However, we can regain the completeness guarantee with a simple, minor change to the algorithm! We use the learned models to evaluate the proposed subgoals, and the true environment dynamics only for simulating the consequences of the low-level actions and validating the found trajectories at the end of the search. Due to this validation and use of low-level actions, the search is complete, and the number of low-level environment steps is reduced to only N_ACTIONS (=4 in our experiments) * N_EXPANSIONS + a small overhead of validating the found trajectories ($\approx 50$ in Sokoban), which outperforms even AdaSubS! We ran a preliminary experiment on Sokoban and the results indicate that the proposed change does not weaken the results in Table 1 in terms of node expansions. We also evaluated the proposed approach on Box-World, where the learned model is inaccurate, and confirmed that the solution percentage is 100, while the number of low-level environment steps is minimal. We hope that you agree that this modification strengthens the paper considerably.
> > > > >
> > > > > We will add the complete outcomes of the experiments and the proposed comparison to AdaSubS in the next version of the paper as additional results.

---

> > > > > > ### Comment · Reviewer_v7TG · 2023-08-19
> > > > > >
> > > > > > > However, this calculation assumes that the low-level expansions are evenly distributed between the start and end of the search.
> > > > > >
> > > > > > At first, I thought it's not true, but after rethinking I agreee. Thus, I think it would be also beneficial to analyze the distribution of using low-level expansions wrt the stage of the search (i.e. on the y-axis the % of low-level expansions, on the x-axis the number of high-level nodes already expanded in the search). This would also help in understanding how often are the low-level expansions used within the budgets you analyze. I think that such analysis as you presented in general would be very interesting and helpful to the readers. I suggest including it in the paper or appendix.
> > > > > >
> > > > > > > almost every (>98 %) low-level expansion is performed early in the search (before 10 expansions), including the problem instances which the high-level search can solve without relying on low-level actions.
> > > > > >
> > > > > > Just to clarify, do I understand correctly that Figure 7a shows the percentage of low-level actions _in the final solution paths_ and 7b the percentage of low-level expansions _in the full search tree_?
> > > > > >
> > > > > > > Note that the inefficiency in low-level environment steps is primarily a property of HIPS
> > > > > >
> > > > > > Yes, I acknowledge that. It doesn't impact the completeness guarantees, only the practical efficiency report.
> > > > > >
> > > > > > > However, using a learned dynamics model allows us to significantly reduce the number of low-level environment steps.
> > > > > >
> > > > > > Yes, that's true. However, it doesn't reduce the complexity of the search itself. I suggest adding a short remark about the budget in the paper, and preferably a full-budget comparison somewhere, to be on the safe side, as you propose.
> > > > > >
> > > > > > > However, we can regain the completeness guarantee with a simple, minor change to the algorithm!
> > > > > >
> > > > > > I like this idea, I suggest putting it in the paper since it's elegant and fits well your experiments.

---

> > > > > > > ### Author Response · Authors · 2023-08-20
> > > > > > >
> > > > > > > Thank you for the time invested, comments and suggestions. To summarize, we will follow all the suggestions. Please, see below for more details.
> > > > > > >
> > > > > > > >At first, I thought it's not true, but after rethinking I agreee. Thus, I think it would be also beneficial to analyze the distribution of using low-level expansions wrt the stage of the search (i.e. on the y-axis the % of low-level expansions, on the x-axis the number of high-level nodes already expanded in the search). This would also help in understanding how often are the low-level expansions used within the budgets you analyze. I think that such analysis as you presented in general would be very interesting and helpful to the readers. I suggest including it in the paper or appendix.
> > > > > > >
> > > > > > > We agree, we will add the analysis and the plot you suggested.
> > > > > > >
> > > > > > > >Just to clarify, do I understand correctly that Figure 7a shows the percentage of low-level actions in the final solution paths and 7b the percentage of low-level expansions in the full search tree?
> > > > > > >
> > > > > > > Yes, this is true.
> > > > > > >
> > > > > > > >Yes, that's true. However, it doesn't reduce the complexity of the search itself. I suggest adding a short remark about the budget in the paper, and preferably a full-budget comparison somewhere, to be on the safe side, as you propose.
> > > > > > >
> > > > > > > We will do this by adding the budget remark and a full-budget comparison.
> > > > > > >
> > > > > > > >I like this idea, I suggest putting it in the paper since it's elegant and fits well your experiments.
> > > > > > >
> > > > > > > We will add this to the paper.

---

> > > > > > > > ### Comment · Reviewer_v7TG · 2023-08-20
> > > > > > > >
> > > > > > > > Thank you for the discussion, clarifications and proposed changes. I acknowledge that by increasing my score.

---

### Official Review · Reviewer_nqzu · 2023-07-06

**Soundness:** 3 good
**Presentation:** 3 good
**Contribution:** 2 fair
**Rating:** 5
**Confidence:** 3

**Summary:**

The paper considers solving complex planning problems with discrete action spaces and develops a novel hybrid search scheme that combines high-level sub-goal oriented search (aka hierarchical planning) with a complete low-level search scheme. The latter embodies a classical exhaustive search scheme that only considers low-level actions. The proposed approach is applied to an existing sub-goal oriented planning system called HIPS (Hierarchical Imitation Planning with Search). In contrast to the existing approaches, including HIPS, the new system is guaranteed to be complete, namely it will find a solution if one exists. Furthermore, the proposed enhanced HIPS is evaluated on four planning benchmarks which were also considered in previous work. The results demonstrate clearly the performance of the proposed approach compared with the baseline HIPS system as well as with strong existing offline reinforcement learning algorithms.


**Strengths:**

The paper is fairly well written and organised. The quality of the presentation is overall fairly good. The results are presented in a relatively clear manner so it's fairly easy to grasp the big picture.


**Weaknesses:**

My only concern is that the proposed approach looks fairly incremental compared with the existing work on HIPS. The main novelty seems to consist in adding the behaviour cloning policy to select low-level actions.



**Questions:**

1. The sliding puzzle and to some extent the box-world problems are considered to be fairly easy to solve by classical AI planners using some version of A* search. I was wondering how does the proposed HIPS enhancement compare with classical planners on this domain.



**Limitations:**

I think the limitations of the proposed method are discussed fairly clearly in the paper.

---

> ### Author Rebuttal · Authors · 2023-08-09
>
> We thank the reviewer for their insightful comments and observations.
>
> > The main novelty seems to consist in adding the behaviour cloning policy to select low-level actions.
>
> We want to emphasize that not only do we add a new behavior cloning policy to efficiently solve a known problem of subgoal search methods, but we also analyze the impact it will have on the results, derive a new heuristic rule for efficiently using it in search (which is particularly important, see Table 4) and confirm that the empirical results match the expected ones. High-level search has been recognized as a promising research direction, as kSubS was published at NeurIPS 2021, AdaSubS at ICLR 2023 (Notable top-5 % = Oral), and HIPS at ICML 2023, so improving on these methods is relevant. Furthermore, we show that using the low-level actions not only guarantees completeness but can also improve the search performance on instances solvable by high-level search, which is a non-trivial result. Although we evaluated our approach on HIPS due to its strong performance, our framework can also be applied to other subgoal search methods such as kSubS and AdaSubS. Finally, we show promising OOD generalization capabilities, which are missing in many learning algorithms. The promising OOD results also open doors for new applications such as curriculum learning and the multi-task setting.
>
> > The sliding puzzle and to some extent the box-world problems are considered to be fairly easy to solve by classical AI planners using some version of A* search. I was wondering how does the proposed HIPS enhancement compare with classical planners on this domain.
>
> Defining a suitable heuristic for the Box-World problem (without prior knowledge about the solution path) is a highly non-trivial problem, which hampers our ability to apply the A* algorithm. If we naively apply Dijkstra's algorithm to it, we're bottlenecked by RAM before a solution is discovered (happens at ~700k node expansions). If we assume access to prior knowledge, a reasonable heuristic is to count the number of collected keys (which does not separate between distractors and correct ones) and subtract that from the goal length (requiring prior knowledge). This heuristic does not help us to reliably discover solutions (see Table 12 in the global rebuttal pdf) even if we perform WA* with W=10, and in most of the failure cases, we run out of RAM before a solution is discovered.
>
> For STP, note that the prior work in the planning domain has focused on the easier 4x4 variant, whereas we work on the 5x5 problem, which is considered significantly harder (see [1], pp. 71). Nevertheless, we can still use the Manhattan distance as a heuristic for A*. We experimented with a limit of 100,000 expansions (note that HIPS-$\varepsilon$ had a 69.5 % solution rate with 100 expansions and 93.8 % at 200). A* had a solution percentage of 0. With WA* and W=2, the solution % at 100,000 expansions was 10.2 %, which is significantly worse with 10^3 times more node expansions. For W=5, where the heuristic is used very greedily, the solution rate is 91.0 % at 100,000 expansions (see Table 12 in the rebuttal pdf). Even then, it is significantly inferior to HIPS-$\varepsilon$ in terms of expansions and requires applying prior knowledge for defining the heuristic, whereas HIPS-$\varepsilon$ does not assume any prior knowledge except the ability to recognize terminal states upon entering them. Finally, note that, for example, the HIPS paper [2] specifically used subgoal-based A* to generate the demonstrations for STP and Box-World because of how expensive the demonstration generation was with standard A*.
>
> [1] Russell, S. & Norvig, P. Artificial Intelligence: A Modern Approach. 3rd edition.
>
> [2] Kujanpää, K., Pajarinen, J., & Ilin, A. (2023). Hierarchical Imitation Learning with Vector Quantized Models. ICML 2023.

---

### Author Rebuttal · Authors · 2023-08-09

We want to thank all reviewers for taking the time to review our work and give feedback. Your insightful comments and observations are vital for improving the paper. We are grateful to the reviewers for appreciating the empirical results (all reviewers), the intuitive main idea (v7TG), clear motivation (jgQs), the broad evaluation and insightful comments (jgQs), the OOD application (v7TG, jgQs), and the presentation and writing (all reviewers).

To address the questions of the reviewers, we attached a pdf with results from the following additional experiments:
1. OOD experiments on Sokoban with six boxes, when the model has been trained on Sokoban with four boxes (Figure 6).
2. Demonstrating how the number of low-level expansions made by the search and the number of low-level actions in the returned solutions are affected by the value of $\varepsilon$ in STP (Figure 7).
3. Comparing HIPS-$\varepsilon$ to classical planning algorithms in terms of node expansions, highlighting the difficulty of the problems HIPS-$\varepsilon$ is capable of solving (Table 12).

We plan to add these to the final version to strengthen the paper even further. Furthermore, in the final version, we want to motivate why the completeness property is relevant (v7TG), improve the related work section to clarify the relation of our work to the prior work (i9rx), and make other adjustments based on the reviewer feedback.

---

### Decision · Program_Chairs · 2023-09-21

**Decision:**

Accept (poster)

**Comment:**

The reviewers were in general agreement that the paper made a solid contribution. Please take the critical feedback into account when preparing the camera ready paper.